# Historical overview and geographical distribution of neglected tropical diseases amenable to preventive chemotherapy in the Republic of the Congo: A systematic review

Joseph A. Ngatse[1,2]*, Gilbert Ndziessi[1], François Missamou[3], Rodrigue Kinouani[4], Marlhand Hemilembolo[3], Sébastien D. Pion[2], Kirsten A. Bork[2], Ange A. Abena[1☉], Michel Boussinesq[2☉], Cédric B. Chesnais[2☉]*

1 Faculté des Sciences de la Santé, Université Marien NGOUABI, Brazzaville, République du Congo, 2 UMI233, Institut de Recherche pour le Développement (IRD)-INSERM U1175-Université de Montpellier, Montpellier, France, 3 Programme National de Lutte contre l'Onchocercose, Brazzaville, République du Congo, 4 Centre de Recherche Géographique et de Production Cartographique, Brazzaville, République du Congo

☉ These authors contributed equally to this work.
* josaxel@yahoo.fr (JAN); cedric.chesnais@ird.fr (CBC)

## Abstract

### Background

Neglected Tropical Diseases amenable to Preventive Chemotherapy (PC-NTDs) affect the poorest populations around the world, especially in Africa. Scientific information on the distribution and level of endemicity of these diseases in the Republic of the Congo (RoC) is scarce in the published literature. We sought to collect all available epidemiological data on PC-NTDs in the RoC to document the historical and current situation and identify challenges in reaching the elimination of NTDs.

### Methods

We searched Medline and Horizon databases for studies published until to July 4$^{th}$, 2019, on onchocerciasis, lymphatic filariasis, soil-transmitted helminth infections, schistosomiasis, and trachoma in the RoC. Unpublished reports were also reviewed. We included all epidemiological studies containing community data and excluded case reports. Location, prevalence data, and dates of the studies were extracted.

### Principal findings

We identified 933 records, of which 56 met the inclusion criteria. The articles published before 1960 mainly concerned onchocerciasis and schistosomiasis. Despite a low number over the studied period, since 2005 there has been a steady increase in the number of publications. Most of the studies were cross-sectional and conducted in the general population. Trachoma is endemic in the Sangha and Likouala departments (prevalence of trachomatous inflammation-follicular > 5% in some villages), and further mapping is essential to properly

**Data Availability Statement:** All relevant data are within the manuscript and its Supporting information files.

**Funding:** JAN received a funding through the fellowship offered by the "Coordination Organization for the Control of Endemics in Central Africa (in French, OCEAC)", based on the financial cooperation between the Economic and Monetary Community of Central Africa (in French, CEMAC) and the German Federal Ministry for Economic Cooperation and Development (BMZ) and administered by the "Kreditanstalt für Wiederaufbau (KfW)". The funders had no role in study design, data collection and analysis, decision to publish, or preparation of the manuscript. GN, FM, RK, MH, SDP, KAB, AAA, MB, and CBC received no specific funding for this work.

**Competing interests:** The authors claim to have no conflict of interest.

assess the burden of this disease in the country. While the prevalence of soil-transmitted helminths is still high (over 20%) in a large part of Congo, cases of lymphatic filariasis (based on *Wuchereria bancrofti* antigenaemia and/or microfilaraemia) and onchocerciasis are becoming rare and very focused. To achieve the elimination of PC-NTDs, further intervention is required.

## Conclusions

Except for trachoma, whose epidemiological situation should be better evaluated, PC-NTDs are endemic in the RoC, and actions to control them have been taken by health authorities. To eliminate PC-NTDs, which are still present in some locations, new mapping surveys are needed, and increased investment in scientific research should be encouraged in the country.

### Author summary

For many years, the Republic of the Congo has implemented control programs to combat neglected tropical diseases that cause severe disabilities. By tracing the past and recent distribution of these diseases through the analysis of epidemiological studies, we show that most remaining NTDs are located in defined foci of infection, maintained depending on ecology and lifestyle habits. However, the small number of recent studies limits the production of new knowledge, which would be useful for a better understanding of epidemiological patterns and to accelerate NTD elimination.

## Introduction

Neglected Tropical Diseases (NTDs) are a group of primarily communicable and tropical diseases affecting rural populations in resource-limited countries. Worldwide it is estimated that more than 1 billion people are affected by at least one NTD [1]. The NTD concept emerged from an international workshop organized in 2003 in Berlin (Germany) by the World Health Organization (WHO) and German institutions [2] with the initial intention of intensifying the control of these diseases. A major recent development has been implementing an integrated approach that simultaneously fights different NTDs to efficiently control and eliminate these diseases closely related to poverty [3, 4].

The clinical signs of NTD infections cause significant disability, discrimination, and stigma [5, 6]. For example, onchocerciasis and trachoma can lead to blindness, lymphatic filariasis (LF) to elephantiasis (major lymphedema, usually of lower limbs), and Buruli ulcer can reduce mobility and lead to skin cancer [6–8]. Although NTDs are mainly disabling, some of them, such as rabies, Human African Trypanosomiasis (HAT), or snakebite envenoming, can also lead to death if not diagnosed and promptly treated [7].

NTDs are divided into two groups. First are NTDs amenable to preventive chemotherapy (PC-NTDs), such as mass drug administration (MDA). This group includes five diseases (or groups of diseases): LF, onchocerciasis, infections with soil-transmitted helminths (STH), schistosomiasis, and trachoma. Second are those NTDs combated by active case detection and individual treatment (case management [CM]-NTDs). This group includes four helminthiases (dracunculiasis, taeniasis/cysticercosis, foodborne trematodiases, and echinococcosis), three

protozoan diseases (HAT, Chagas disease, and leishmaniasis), three bacterial diseases (leprosy, Buruli ulcer, and yaws), two viral diseases (rabies and dengue/chikungunya), as well as snake-bite envenoming, deep mycoses, and scabies and other ectoparasitic infections [9].

In 2017, more than 140 million new cases of NTDs were reported worldwide, adding to the number of ongoing infections, which now totals more than one billion people living mainly in Africa, Asia, and Latin America [10]. The same year, NTDs caused the death of approximately 100,000 individuals, mainly by dengue (40.5%, 40,500 deaths), rabies (11.7%, 11,700 deaths), and schistosomiasis (8.8%, 8,800 deaths) [11].

The Republic of the Congo (RoC) is located in Central Africa, covering 342,000 km$^2$ with more than 5 million inhabitants [12]. The climatic, hydrographic, and landcover characteristics of the country, with forests covering nearly 65% of the national territory, the agricultural and hunting activities of the rural population, and the weakness of the country's health system are all factors contributing to the continued presence of many NTDs [12]. The 2013 Global Burden of Disease (GBD) Study found that worldwide, the RoC had the eighth highest prevalence of HAT (2.7 per 100,000 inhabitants) and the 10th highest prevalence of LF (7.0/100,000) and ascariasis (32.7/100,000) [13]. Within the country, schistosomiasis was the seventh greatest cause of years of life lived with disability [14].

Historically, the RoC has always paid particular attention to the fight against endemic infectious diseases. Immediately after gaining independence from the French colonial sovereignty in 1960, the Congolese government created, within the Ministry of Health, a major endemic diseases (MED) service, with objectives of combating endemic diseases similar to those of the colonial "Services Général d'Hygiène Mobile et de Prophylaxie" (one in French West Africa and one in French Equatorial Africa). This Service is now named "Operational Sector" with offices in each of the RoC's twelve departments (administrative divisions). Regionally, the Organization Coordination and Cooperation for the fight against the Grandes Endémies in Africa Central, OCCGEAC was established in 1963 in Yaounde to coordinate the fight against MEDs in the central African countries of Cameroon, Gabon, Central African Republic (CAR), Chad, and RoC. In 1965, the OCCGEAC became the Organization for the Coordination of the Fight Against Endemic Diseases in Central Africa (OCEAC, in French the Organisation de Coordination pour la lutte contre les Endémies en Afrique Centrale) [15]. The diseases targeted by the OCEAC included most of the NTDs (both PC- and CM-NTDs).

Beginning in the 1980s, following the reorganization of the health system in the RoC, specific programs were set up to more effectively combat each NTD. Programs for HAT and leprosy were established in 1980, onchocerciasis and LF in 1984, schistosomiasis and STH in 1986, Buruli ulcer in 2005, and yaws in 2006. After the many years since their launch, it is important to review the information regarding these control activities. In this context, we conducted a monograph-type historical review on the five PC-NTDs in the RoC, for which a brief disease description is presented in the results section, focusing on prevalence rates measured in surveys of the general population. The aims were to (i) inform the international community about the past and present epidemiological status of these diseases, (ii) provide useful information to control programs on the historic endemic infection foci, (iii) provide a bibliographical base for researchers interested in NTDs, and (iv) show the epidemiological trends regarding these diseases in the RoC.

## Methods

### Research strategy

The exhaustive litterature search was conducted from July 4, 2019, to July 4, 2019 and identified eligible references published from 1914 to 2019. The search used references indexed in

MedLine and the Horizon database of the French Institut de Recherche pour le Développement (IRD) (https://horizon.documentation.ird.fr/exl-php/cadcgp.php?CMD=CHERCHE&query=1&MODELE=vues/horizon/accueil.html&AUTH=1). We also searched relevant references listed in the articles identified in the MedLine and Horizon databases, and control program and unpublished scientific reports.

### Selection procedure

Searches were done separately for each of the five PC-NTDs as listed by the WHO [16] and included articles and reports written in French or English. The general search query was "Name of the disease sought AND Congo." We chose as inclusion criteria, community-based epidemiological studies conducted in the RoC reporting prevalence values for at least one NTD. For schistosomiasis, studies presenting both malacological and epidemiological data were retained, but only epidemiological data were extracted. We also included data presented in unpublished reports of the national control/elimination programs or doctoral theses.

We excluded duplicates published in different journals or presented both in reports and in publications. In addition, we excluded non PC-NTDs-related studies, clinical case reports, animal studies, and epidemiological studies reporting data other than prevalence data (diagnostic performances, associated risk factors, etc.).

### Use of data

Once the screening was finalized, we extracted information on the year of the study, study site location (including the village or district and the geographic coordinates), number of individuals surveyed, prevalence values, and the epidemiological index used (e.g., the prevalence of nodules (PNod) or skin microfilariae (PMF) for onchocerciasis). When the geographic coordinates were missing, village positions were determined using maps available at the "Centre de Recherche Géographique et de Production Cartographique" (CERGEC) in Brazzaville. The maps presented in this review were produced using MapInfo v8.5 software.

For each PC-NTD, we present (i) a summary of the clinical presentation and diagnostic methods, (ii) a history of epidemiological surveys conducted, and (iii) data on the epidemiological surveys and MDA organized by the Programme National de Lutte contre l'Onchocercose (PNLO, National Onchocerciasis Control Program) which coordinates the activities for the five PC-NTDs. In addition, for onchocerciasis, information on studies implemented by the African Programme for Onchocerciasis Control (APOC, 1995–2015) is also given. To facilitate the reading of the abbreviations, a list was provided (S1 Appendix). We also provided the preferred reporting items for systematic reviews and meta-analysis (S2 Appendix), including pages from each part of our literature review.

In the results, we define "administrative district" (AD) as the territorial administrative subdivision immediately following the department. "Health district" (HD) corresponds to the geographic and administrative subdivision of the health system consisting of a referral hospital (or district hospital) surrounded by a network of public (health posts and centers) and private (medical and social centers, medical offices, clinics, etc.) health facilities [17, 18]. In the RoC, the HD corresponds to either an AD, an arrondissement or a grouping of ADs or arrondissements.

## Results

### Source data selection

We identified 933 documents (articles and unpublished reports) in our initial search, including one duplicate. We discarded an additional 876 documents that did not meet the inclusion

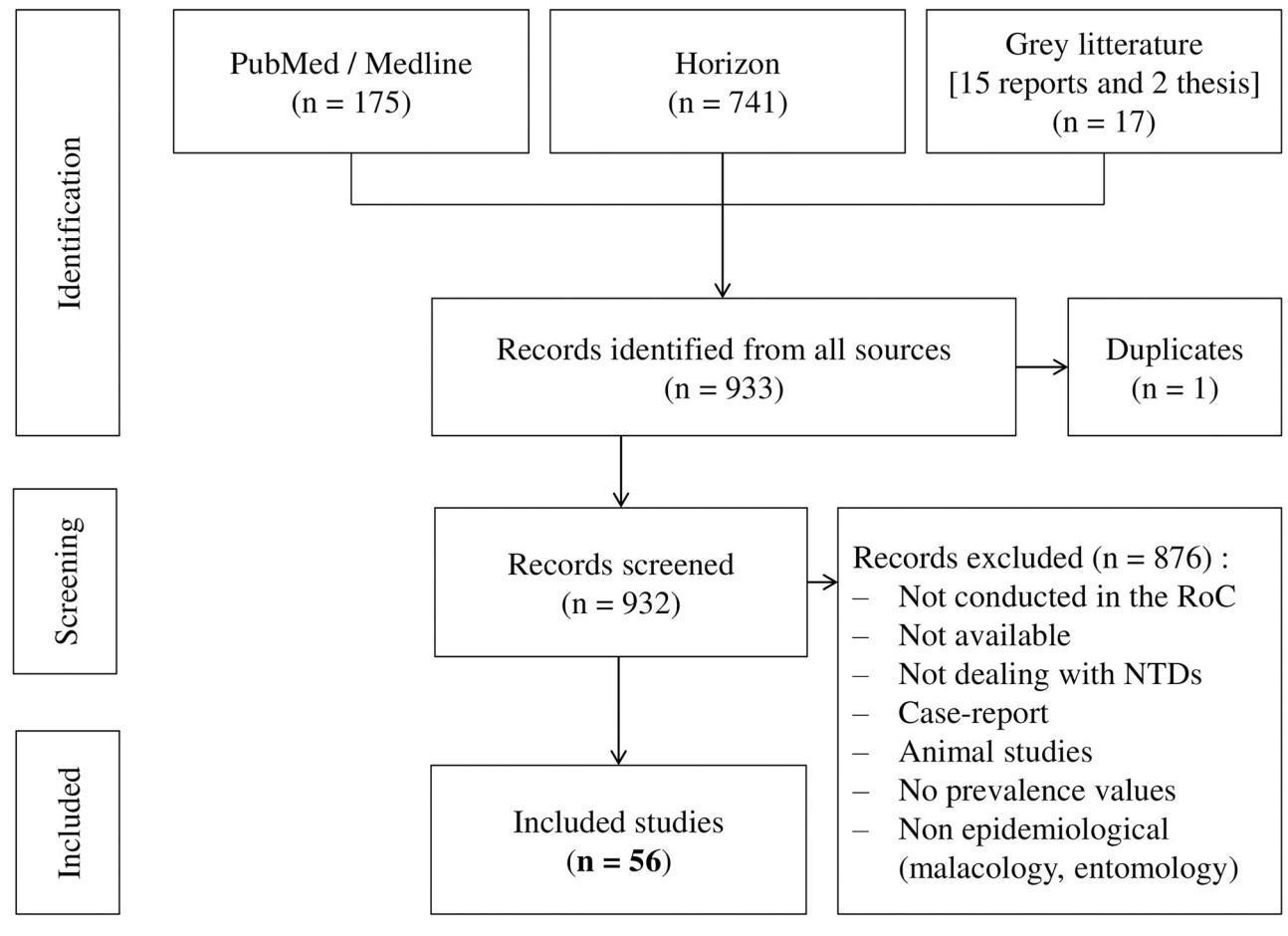

**Fig 1. Flowchart of selected studies.**

criteria based on title and abstract (e.g., reports from the Democratic Republic of Congo or clinical case reports). Our systematic review is of the remaining 56 documents, including 38 articles and 15 reports, which met the inclusion criteria (Fig 1) and for which all references could be identified. The year of publication ranged from 1920 to 2019.

## Number of included documents according to the publication period and the type of NTDs

Fig 2 presents the number of documents included in this review according to the publication period and disease. It shows that most focused on onchocerciasis and schistosomiasis. The sharp decrease in the number of studies between 1996 and 2010 is probably due to the 1997–1998 civil war and the following years of political instability.

## History of national control programs for PC-NTDs

There are two main control programs for PC-NTDs in the RoC: the PNLO mentioned above and the Programme National de Lutte contre la Schistosomiase (PNLSCH, National Program for Schistosomiasis Control). Due to a lack of personnel ensuring the sound technical management of the PNLSCH, its activities were, for a long time, managed by the PNLO. From 1984 to 1991, the PNLO conducted epidemiological and entomological surveys to identify

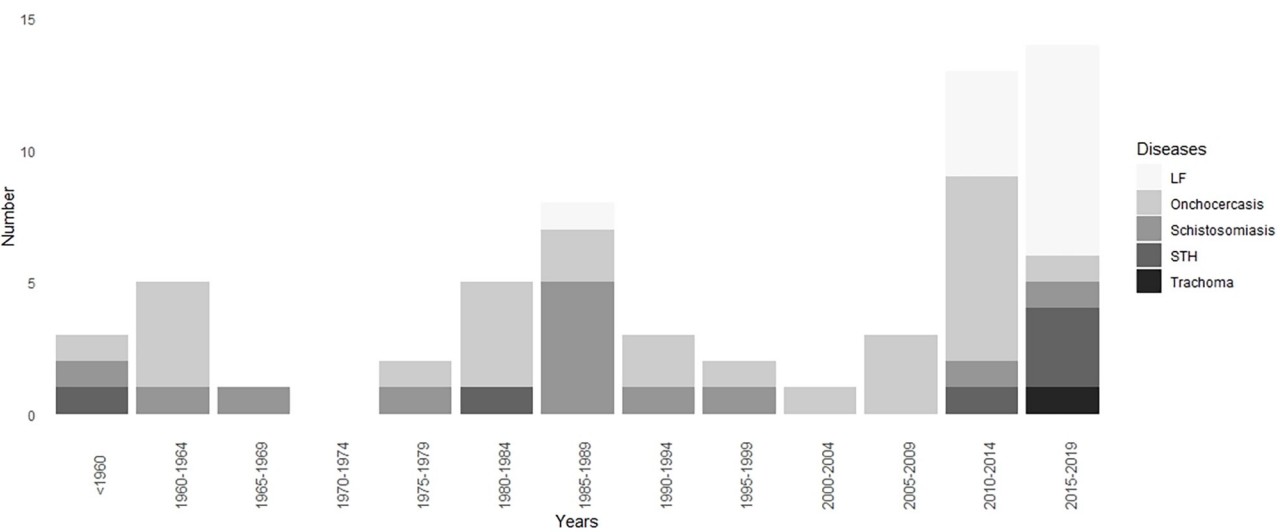

**Fig 2. History of the number of articles and reports published for each PC-NTD (before 1960 then during each 5-year period).**

onchocerciasis foci. From 1992 to 2000, it implemented MDAs in identified foci, using mobile teams with health personnel visiting different villages. From 2001 onwards, the PNLO (supported by APOC until its closure in 2015) continued the MDAs, using the community-directed treatment with ivermectin (CDTI) strategy, i.e., with resident(s) responsible for treating their village. In addition, it conducted studies to assess the levels of infection in the onchocerciasis foci under treatment and in areas not previously surveyed. The PNLO was also responsible for control activities against LF and STH and conducted a 2015 trachoma survey.

A summary of key events regarding the control of PC-NTDs in the RoC is presented in Fig 3. Several partners supported the PNLO to achieve its objectives. The first was the WHO through APOC, and then, from 2016, the Expanded Special Project for Elimination of Neglected Tropical Diseases (WHO/ESPEN). Other key partners were non-governmental development organizations (NGDOs), including the Organisation pour la Prévention de la

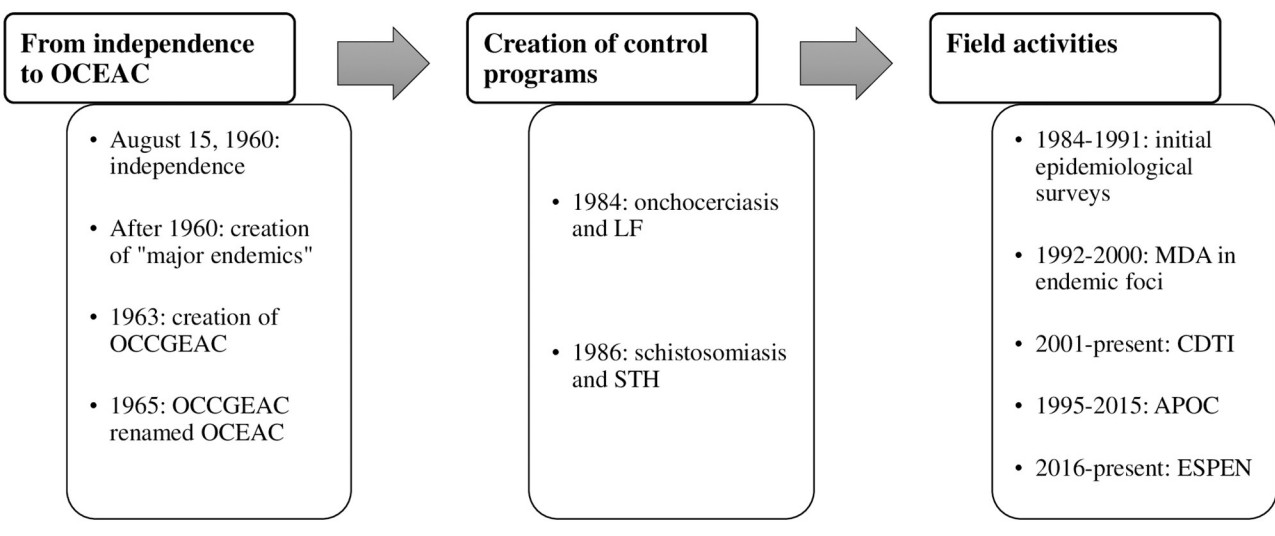

**Fig 3. History of national control programs for PC-NTDs.**

Cécité (OPC, Organization for the prevention of blindness) and Sightsavers; the Mectizan Donation Program; and, since 2019, an NTD program coordinated by the OCEAC and funded by the German Development Bank (Kreditanstalt für Wiederaufbau, KfW).

## Onchocerciasis

**Clinical presentation and diagnostic methods.** Onchocerciasis is a parasitic disease caused by the filarial worm *Onchocerca volvulus* and transmitted from human to human by *Simulium* blackflies that breed in fast-flowing rivers. Besides the subcutaneous nodules, which contain adult worms, its main manifestations are cutaneous and ocular. The presence of larval stages (microfilariae, mf) in the dermis causes itching, which can be severe, and different types of lesions: acute papular, chronic papular or lichenified onchodermatitis and, in advanced cases, skin atrophy and a typical depigmentation over the anterior shin ("leopard skin"). The presence of mf in the anterior and posterior segments of the eye induces lesions which can cause visual impairment and blindness (onchocerciasis is also known as "river blindness") [19]. While visual impairment has been associated with excess mortality [20, 21], individuals with a high microfilarial load may also have a decreased life expectancy [22]. In addition, the recent demonstration of a temporal relationship between onchocerciasis and epilepsy (highly parasitized children being at risk for the latter) [23, 24] confirmed observations made in the 1930s in Mexico [25].

The standard diagnostic method is the demonstration of *O. volvulus* mf in small skin biopsies (skin snips) taken from the iliac crest area with a corneoscleral punch. The endemicity level of the disease is usually measured by the PMF (also called "microfilarial index") or the PNod (also wrongly called "cystic index") in the population [26]. APOC defined four levels of endemicity according to the PNod in subjects aged ≥20 years: non-endemic, hypoendemic, mesoendemic, and hyperendemic, for PNod <5%, 5–20%, 20–40%, and >40%, respectively [27].

**History of epidemiological surveys.** The first cases of onchocerciasis in the RoC were reported by Lebœuf (quoted by Ouzilleau et al. [28]) in 1919, in villages of the Djoué River valley (which flows into the Congo River just west of Brazzaville). The presence of the disease was confirmed two years later in many villages of the Pool region, particularly those located near the Djoué and Foulakari Rivers (the latter also flows into the Congo with waterfalls at the confluence point) [28]. The primary blackfly vector in the RoC belongs to the *Simulium damnosum* s.l. complex. It was first reported in 1943 in the Louvisi River, a tributary of the Niari River [29]. Cytotaxonomic studies showed that the species transmitting *O. volvulus* in the Brazzaville area is *S. squamosum* [30].

A series of cross-sectional surveys were conducted in 1960–1961 in the Bouenza, Plateaux, and Pool departments to assess the prevalence of onchocercal nodules, visual impairment, and skin mf. Some of these studies included adults and children and others only adults. Two techniques were used to search for skin mf: the scarification technique and the examination of skin biopsies taken from the trochanter region with surgical scissors. The surveys showed that onchocerciasis was present (at very low prevalence as presented in the Table 1) in the Madingou area (Bouenza department) and the Abala and Gamboma areas (Plateaux department) [31]. In the Pool department, the surveys covered the Boko subprefecture (1331 subjects from 19 villages, mainly adults, examined in January-March 1961), the Mindouli subprefecture (1519 adult males from 95 villages examined in September 1961), and the Kindamba-Mayama subprefecture (395 children and 1424 adults from 72 villages of the Djoué and Djouéké River valleys examined in October-November 1961) [32–34]. The study in the sub-prefecture of Boko reported PNod of 60.7% (476/784) in the Bacongo canton and 41.8% (148/354) in the

**Table 1. Summary of the included epidemiological studies and reports for onchocerciasis (N = 27).**

| Study | Year of survey | Departments | Villages | Design | N surveyed | Main results$ |
|---|---|---|---|---|---|---|
| **Therapeutic assessment surveys** | | | | | | |
| APOC report, 2013 [57] | 2011 | Bouenza | Ten villages | Cross-sectional | *Not available* | Decrease in the PNod from 16.5–71.4% in 2004 to 0.6–6.7% in 2011 |
| **Mass drug administration therapeutic coverage (WER reports)** | | | | | | |
| 2014 [64] | 2013 | All | Targeted villages | Cross-sectional | 1,427,670 | TC by PNLO: 48.2% |
| 2013 [63] | 2012 | Meso- & hyper-endemic departements | Targeted villages | Cross-sectional | 848,286 | TC by PNLO: 81.2% |
| 2012 [62] | 2011 | | | | 844,984 | TC by PNLO: 81.2% |
| 2010 [61] | 2009 | | | | 764,915 | TC by PNLO: 80.7% |
| 2009 [60] | 2008 | | | | 629,030 | TC by PNLO: 76.1% |
| 2008 [59] | 2007 | | | | 609,925 | TC by PNLO: 73.6% |
| **Prevalence surveys** | | | | | | |
| Niama et al., 2019 [58] | 2018 | Kouilou, Niari | Kouilou-Niari River Basin | Comparative | 2211 | Reduction in the PMF from 50.4% in 2004 to 11.4% in 2018, due to MDA of IVM. |
| Zoure et al., 2014 [45] | 2011 | National level | Geostatistical analysis | Geostatistical analysis | | High-risk villages are mainly in Brazzaville and Pool departments |
| Noma et al., 2014 [44] | 2011 | National level | 384 villages selected according to ecology | Cross-sectional | 13,853 | High-risk villages are mainly in Brazzaville and Pool departments |
| Talani et al., 2005 [37] | 2000 | National level | 94 villages across the country | Cross-sectional | 30–50 | PNod in Pool department: Mayama-Poste (58.3%), Ndzouengue (37.1%) and Bangou-Louholo (34.3%) |
| Noma et al., 2002 [46] | 2001 | National level | Villages selected according to ecology | Cross-sectional | | Onchocerciasis is highly-endemic in southern departments of RoC: Bouenza, Lekoumou, Pool, Brazzaville and Kouilou |
| Talani et al., 1997 [36] | 1992 | Brazzaville | Makelekele | Cross-sectional | 1189 | PMF: 40%, and higher in males (p<0.005). |
| Carme et al., 1993 [26] | *Not available* | Pool | Kibouende, Madibou, Mayama and Mandombe | Cross-sectional | 991 | PMF in subjects >15 years old: Kibouende (2.0%), Sossolo (0%), Madibou (3.9%), Mayama (18.1%), Mandombe: 18.6% |
| Carme et al., 1990 [43] | 1978–1987 | All departments except Sangha | | Literature review | 25 | Northern departments of RoC are non-endemic |
| Kaya et al., 1986 [41] | 1985 | Pool | N'tombo Manyanga | Cross-sectional | 190 | PMF (76.7%)—PNod (52.1%) |
| Mialebama et al., 1986 [35] | 1985 | Pool | Foota, Mantaba, Kimpenga, Bela, Mandombe | Retrospective and prospective | 1106 | Global PMF (77.1%) |
| Yebakima et al., 1980 [42] | 1978 | Brazzaville | Mafouta-Massissia | Cross-sectional | 307 | PMF (42.7%)—PNod (14.9%)—CMFL (8.0 mf/ss) |
| Yebakima et al., 1982 [30] | 1981 | Kouilou | Mayombe forest | Cross-sectional | 236 | PMF (50.4%)—PNod (35.3%)—CMFL (6.2 mf/ss) |
| Carme et al., 1982 [40] | 1981 | Pool | M'payaka, Kibouende | Cross-sectional | 384 | PMF (48.4%)—PNod (21.6%)—CMFL (31.4 mf/ss) |
| Yebakima et al., 1980 [38] | 1975 | Pool | Kinssasa | Cross-sectional | 84 | PMF (67.8%)—PNod (27.4%) |
| Yebakima et al., 1978 [39] | 1977–1978 | Pool | Bangou-Louholo | Cross-sectional | 266 | PMF (40.9%)—PNod (21.0%)—CMFL (22.2 mf/ss) |
| Gilles, 1962a [32] | 1961 | Pool | East zone of Kindamba-Mayama prefecture | Cross-sectional | 1819 (≥15years) | PMF (26.0%)—PNod (30%) |
| Gilles, 1962b [33] | 1961 | Pool | Mindouli | Cross-sectional | 1519 | PMF (59.3%)—PNod (22.6%) PMF among subjects without nodules: 47.4% |
| Gilles, 1961a [34] | 1961 | Pool | 13 villages in Bacongo and 6 villages in Bacongo-Tséké | Cross-sectional | 1331 | **Bacongo**: PNod (60.7%) **Bacongo Tseke**: PNod (41.8%) |

*(Continued)*

 

**Table 1.** (Continued)

| Study | Year of survey | Departments | Villages | Design | N surveyed | Main results$ |
|-------|----------------|-------------|----------|--------|------------|---------------|
| Gilles, 1961b [31] | 1960 | Bouenza, Plateaux | Madingou, Abala and Gamboma | Cross-sectional | 3901 | *Onchocerca volvulus* infestation (per thousand) • Madingou: 7–8 have nodules, 72 are infested • Abala: 21 have nodules, 72 are infested • Gamboma: 16 have nodules, 56 are infested |
| Ouzilleau et al., 1921 [28] | 1921 | Brazzaville | Djoué | Cross-sectional | 27 | 3 cases of onchocerciasis in Mbouni PNod (16.3%) |

$ PNod: prevalence of nodules; PMF: prevalence of skin microfilariae; CMFL: Community Microfilarial Load (expressed as microfilariae per skin snip, mf/ss); WER: Weekly Epidemiological Reports.

Bacongo Tseke canton. The vast majority of nodules were located in the pelvic region. In the sub-prefecture of Mindouli, the PNod was 22.6% (344/1519) and a PMF of 47.4% (557/1175) was found in non-nodule carriers. Finally, in the sub-prefecture of Kindamba-Mayama, the author reported a PNod of 13% (187/1424) and a PMF of 26% (372/1424) in adults, and a PNod of 0.5% (2/395) and a PMF of 0.5% (2/395) in children. These studies did not report any cases of blindness.

Later studies conducted between 1978 and 1992 confirmed the endemicity of onchocerciasis in the Pool and Brazzaville regions, with PMF between 40% and 67.8% in subjects aged ≥14 years living in the villages of Foota, Mantaba, Kimpenga, Bela, Mandombe, Makelekele, Mayama-Poste, Ndzouengue, Kinssasa, Bangou-Louholo, Kibouende, and Ntombo-Manyanga; in the Makélékélé district, in the western part of Brazzaville, the PMF in adults was 65.6%, and the average microfilarial density was 29.3 mf/skin biopsy (mf were counted after 30 minutes incubation in saline) [26, 35–42] (See S3 Appendix for the reference 35). These surveys revealed that an important suburban/urban focus of onchocerciasis existed in Brazzaville. A review published in 1990 reports that onchocerciasis was not endemic in the northern departments of the RoC, particularly the Likouala (Impfondo district), Cuvette (Mossaka and Oyo districts), and Plateaux (Lekana and Abala districts) departments [43].

In the Kouilou department (the department covering the RoC's coastline), a survey was conducted in 1981 in five villages located in the Mayombe forest. This study included 236 subjects and PMFs of 50.7, 84.2, and 47.0% were reported in the three villages located near the Kouilou River (Manzi, Sunda, and Camp MAB, respectively); conversely, none of the residents of the villages of Dimonika and Makaba was found infected with *O. volvulus* [30].

**Epidemiological surveys and MDAs organized by APOC.** From its launching in 1995, the APOC initiated large-scale surveys in all APOC countries to delineate areas where onchocerciasis was meso- or hyper-endemic, i.e., where MDA of IVM had to be implemented to control the disease. These surveys were based on the examination of 30 to 50 adult males in selected villages to estimate the proportion of subcutaneous nodule carriers (Rapid Epidemiological Assessment, REA). The entire exercise, called Rapid Epidemiological Mapping of Onchocerciasis (REMO), was conducted in 2001 in the RoC and other APOC countries. A total of 13,853 subjects living in 384 villages located in all departments of the RoC was examined [44–46]. The results obtained from the REMO surveys show that onchocerciasis is mainly present in the southern part of the country, in the Pool, Niari, Lékoumou, Brazzaville, Bouenza, and Kouilou departments. This is consistent with the results obtained during the previous surveys mentioned above. Fig 4 presents both the historical surveys before implementation of APOC and results from REMO on PNod at the national level. A zoomed-in map of

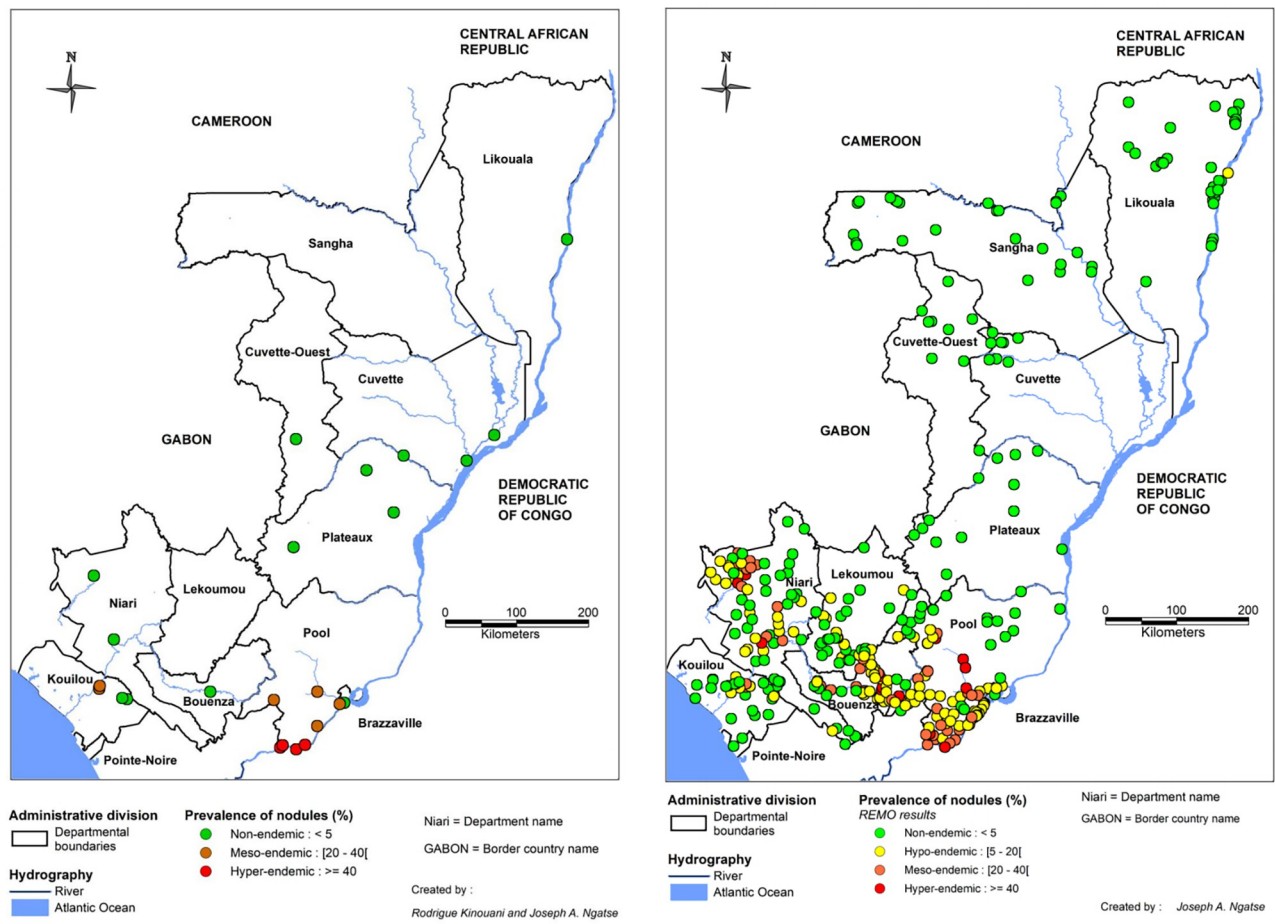

**Fig 4. Distribution of the prevalence of onchocercal nodules in the Republic of the Congo.** The left panel shows the surveys conducted from 1960 to before REMO (2001); the right panel shows the APOC's REMO surveys. The map was created with MapInfo 8.5 (Geographic Information System, http://www.precisely.com). The base layer used of the map was created by the « Laboratoire Population Environnement Développement » (LPD, UMR 151 AMU-IRD) (https://www.lped.fr/-observatoires-societe-environnement-.html) under the supervision of the Ministry of Health and Welfare of the Republic of Congo.

departments with hyper- and meso-endemic villages is presented in the appendix (S4 Appendix).

The PNLO of the RoC faces two specific challenges. The first is that ivermectin (IVM) MDA has to be implemented in Brazzaville and peri-urban areas, where the concept of "community" is difficult to define and where the population is highly mobile. Indeed, individuals can easily move around in these areas and may feel less concerned by onchocerciasis than in rural areas. Hence there might still be a major problem of compliance in these areas. Since compliance is a key factor in eliminating onchocerciasis [47], the use of modern tools such as text messaging would likely help program managers reach the target population.

The second challenge is that in many communities, especially in south-west Congo, onchocerciasis is co-endemic with loiasis (another filarial disease caused by *Loa loa*) and that people with very high densities of *L. loa* mf in the blood can develop severe adverse events (SAEs) after IVM treatment [48–53]. In loiasis-endemic areas where onchocerciasis is mesoendemic or hyperendemic, MDA with IVM is justifiable because the benefit of preventing onchocerciasis-associated morbidity outweighs the risk of post-treatment *Loa*-related SAEs. As the WHO's objective for onchocerciasis has shifted from morbidity control to elimination of infection [1],

hypoendemic areas need to be treated, and alternative treatment strategies (ATS, i.e., different from CDTI) have to be used in those areas where loiasis is co-endemic [54].

**Epidemiological surveys and MDA organized by the PNLO.** The first mass treatment with IVM in the country was organized in 1992 in three villages southwest of Brazzaville (Kombé, Mafouta, and Mantsimou), near the Congo River and its tributary the Djoué, in the present-day Madibou district. The total population of these districts was 7851, of which 2401 individuals were treated out of 5890 eligible individuals (40.8%) [55]. The first APOC-approved CDTI project was carried out in 2001. This project covered meso-hyperendemic communities in five departments: Brazzaville, Pool, Bouenza, Niari, and Kouilou (S5 Appendix contains the names and the population size of the ADs covered in each department). An MDA with IVM was organized in three HD of Brazzaville: Makelekele, Bacongo, and Mfilou-Nga-maba. A total of 191,774 subjects in the target population of 335,903 eligible individuals (58.0%) received treatment [56]. In 2004, a new CDTI project, called "Congo-Extension", was launched to cover 21 additional communities in the Divénié AD (in the Niari department) and one village in the Mayéyé AD, in the southeastern part of the Lékoumou department. These communities had been previously identified in 2001 but not treated.

After a decade of CDTI, the PNLO has conducted surveys to evaluate the prevalence of onchocerciasis nodules in the areas under treatment. These surveys involved villages located in the Bouenza department (August 2011), Pool department (December 2012 at the Congo/DRC border and Djoué and Niari basin sites), and in the Lekoumou, Niari, and Kouilou departments (November-December 2015). In addition, the PNLO evaluated the prevalence of nodules in villages initially defined by REMO as hypo-endemic. These surveys involved the Bouenza, Niari, Lekoumou, and Pool departments (September 2014) and Likouala (November-December 2015). Results are presented in Fig 5.

Fig 6 shows the PMF recorded over the entire national territory before and after 2000. While the study locations were not the same between the two periods, there appears to have been a decrease in prevalence (from hyper- to hypo-endemic) in the Kouilou department, in villages near the Kouilou River.

An epidemiological evaluation of the impact of MDA was conducted in 2011, and, as expected, a significant decrease in the PNod was found in the Bouenza department. Prevalences ranging from 16.5% to 71.4% in 2004 had decreased to 0.6% to 6.7% by 2011 [57]. Finally, a 2018 evaluation of villages in the Kouilou-Niari River basin showed a marked decrease in the PMF, from 50.4% in 2004 to 11.4%, after 14 years of CDTI [58].

During the first three years of IVM MDA, the PNLO targeted 748 communities and achieved geographic coverage (GC) of 56.8%, 62.6%, and 96.1%, for each year from 2001 to 2003, respectively (During APOC, GC was the proportion of identified meso- and hyperendemic communities covered by the CDTI). From 2004 to 2013, in 770 communities targeted by the PNLO, a GC of 99.4% was reached in the first year, which rose to 100% subsequently. Since 2014, the number of targeted communities has steadily increased with a constant GC of 100%, except in 2017 and 2018, when the GC dropped to 85.4% and 96.9%, respectively, because of political unrest in the Pool department. From 2001 to 2019, the PNLO administered 9,387,733 IVM treatments for an average of 494,091 individuals treated every year. The average therapeutic coverage (TC, the ratio of the number of people treated and the total population) was 71.8% during this period. The 80% TC target recommended by the WHO was not reached between 2001 and 2008 nor between 2014 and 2016 (S6 Appendix).

A summary of the epidemiological studies is presented in Table 1. The difference observed, over several years, between the PNLO data (S6 Appendix) and those presented in the Weekly Epidemiological Record (WER) [59–63] is due to the WHOs inclusion of hypo-endemic areas

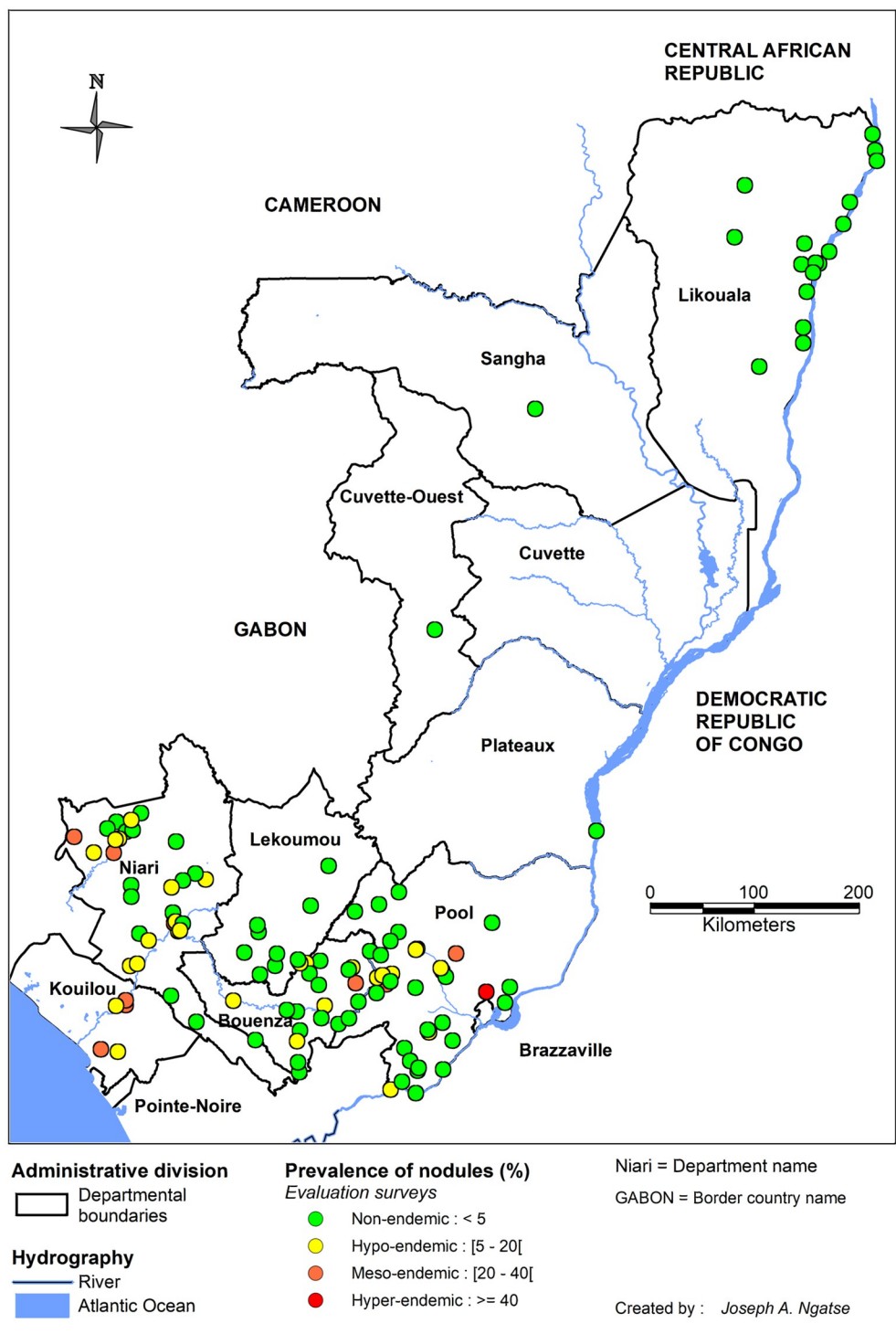

**Fig 5. Onchocercal nodule prevalence during the course of CDTI (2011–2015) in the Republic of the Congo.** The map was created with MapInfo 8.5 (Geographic Information System, http://www.precisely.com). The base layer used of the map was created by the « Laboratoire Population Environnement Développement » (LPD, UMR 151 AMU-IRD) (https://www.lped.fr/-observatoires-societe-environnement-.html) under the supervision of the Ministry of Health and Welfare of the Republic of Congo.

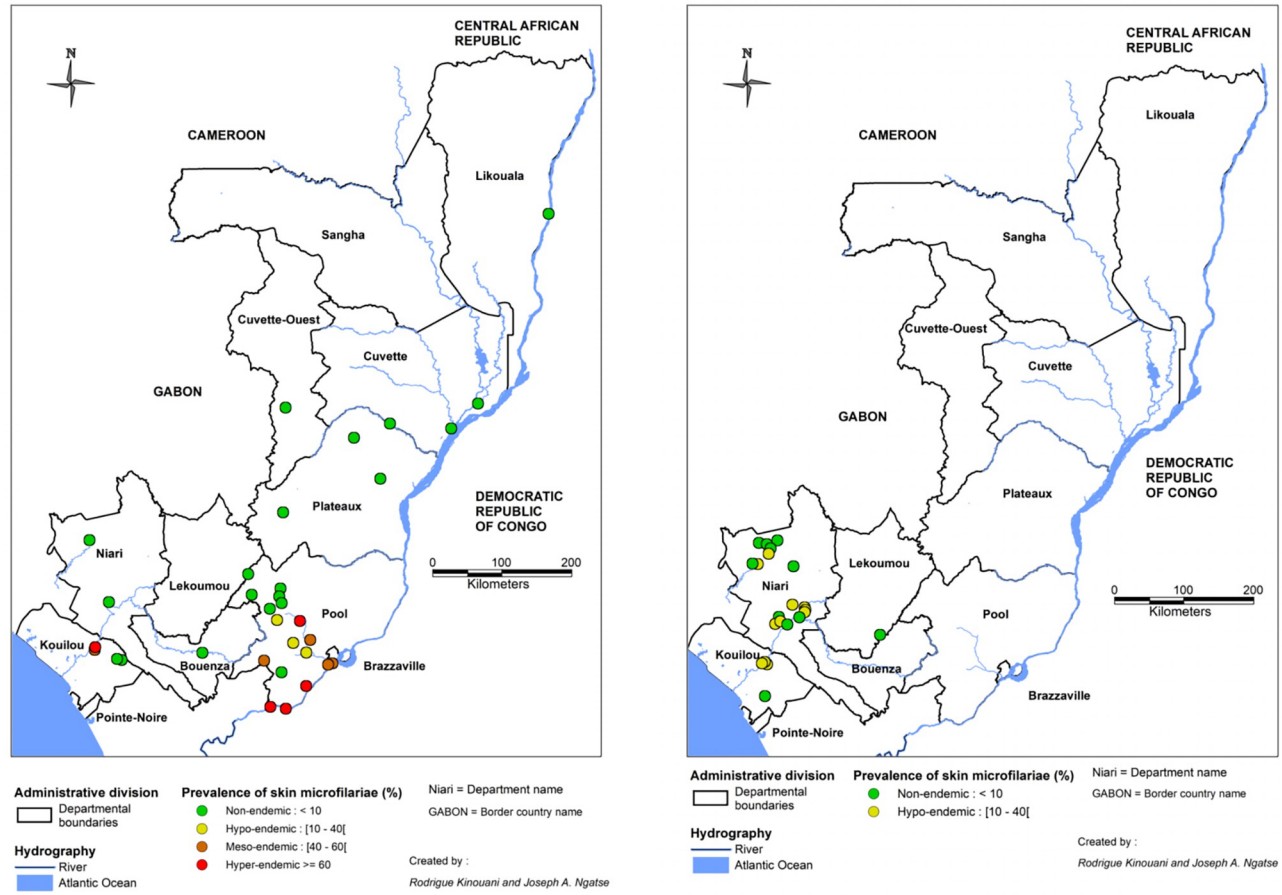

**Fig 6. Prevalence of skin *O. volvulus* microfilariae recorded during surveys in the Republic of the Congo.** The left panel shows the results recorded between 1960 (first survey) and 2000 and the right those of the surveys conducted after 2000. The map was created with MapInfo 8.5 (Geographic Information System, http://www.precisely.com). The base layer used of the map was created by the « Laboratoire Population Environnement Développement » (LPD, UMR 151 AMU-IRD) (https://www.lped.fr/-observatoires-societe-environnement-.html) under the supervision of the Ministry of Health and Welfare of the Republic of Congo.

in the calculation of TC, even though the program did not yet cover these areas for logistic and financial reasons.

## Lymphatic filariasis

**Clinical presentation and diagnostic methods.**    LF is a disease caused by three species of filariae: *Wuchereria bancrofti*, *Brugia malayi*, and *B. timori*, the two latter being restricted to Southeast Asia. The parasites are transmitted by mosquitoes. The main vectors of *W. bancrofti* belong to the genera *Culex* in urban and semi-urban areas, *Anopheles* in rural areas, and *Aedes* in the Pacific Islands [65].

Individuals develop few visible clinical signs at the onset of the infection but can develop chronic and debilitating complications such as hydrocele, lymphedema, and elephantiasis if left untreated. As *W. bancrofti* mf show nocturnal periodicity (i.e., they are found in the peripheral circulation only by night, when mosquitoes are active), infection was classically diagnosed by examining blood smear samples collected at night. Nowadays, infection with *W. bancrofti* is diagnosed by rapid diagnostic tests (RDTs) detecting circulating filarial antigens (CFAs) in the blood, regardless of the time of day [66].

**History of epidemiological surveys.** Little information is available about the presence of LF in the RoC before 2007. In 1914, Ringenbach and Guyomarc'h wrote that "*Microfilaria nocturna*, embryo of Bancroft's filaria, has been reported in the Brazzaville area" [67]. The RoC is not mentioned in the review on LF distribution published by Hawking in 1957 [68]. In 1965, Mouchet et al. [69] wrote that "At Pointe-Noire, where (*W. bancrofti*) transmission is probably very low, one of us found "sausage" stages in one out of 1415 *A. gambiae* dissected and also one out of 2973 *A. funestus*". Hamon et al. [70] wrote only, in another review in 1967, that "annual reports of the RoC Direction of Public Health issued between 1955 and 1961 mention many cases of LF some years, but these reports are hardly usable". No additional information is given by Sasa in 1976 [71] or by Hawking in 1977 [72]. Parasitological surveys conducted in the 1980s failed to find *W. bancrofti* mf, but, as stated by the authors, this could be due to the blood samples having been collected during daytime [73, 74].

**Epidemiological surveys and MDAs organized by the PNLO.** Between December 2007 and February 2008, a nationwide survey was conducted to measure LF endemicity levels in 11 of the 12 departments of the RoC (Pointe-Noire, which became a distinct department in 2003, was not surveyed). This survey, funded by the WHO/AFRO, used the first CFA-detecting RDT: the immunochromatographic card test (ICT). Three communities located in different AD were selected in each department, and at least 100 individuals aged ≥15 years were tested in each community. However, fewer people were tested when the prevalence of antigenemia was high in a community. In total, 3,042 subjects were tested. In five departments (Brazzaville, Plateaux, Cuvette, Lekoumou and Kouilou), all tested individuals were ICT-negative. Conversely, ICT-positive subjects were identified in one AD (Ouesso) of the Sangha department (seroprevalence in the selected community: 12%), in two ADs of the Bouenza department (Mabombo: 33.3%; Mfouati: 1%), three ADs of the Niari department (Kimongo: 37%, Banda: 22%, Nianga: 33%), and one AD of the Pool department (Ignié: 2%). These results are presented in an unpublished report of the PNLO entitled *"Plan directeur de lutte contre les maladies tropicales négligées (MTN) 2018–2022"* (S7 Appendix).

Additional surveys were conducted between July 2010 and April 2013 in the Niari, Bouenza, and Sangha departments (in 24, 13, and 10 villages, respectively) to identify communities where trials could be conducted to evaluate whether MDA with albendazole (ALB) alone could eliminate LF [75]. Volunteers aged ≥5 years were tested by ICT, and those found CFA-positive were invited to be sampled again at night (after 10:00 pm) to prepare standardized thick blood smears (50 μL), which were examined for mf. The number of subjects examined in the Niari, Bouenza, and Sangha departments was 2515, 1311, and 978, respectively. In the Niari department, ICT-positive subjects were found in 10 of the 24 villages surveyed, with prevalence exceeding 3% in three of them (7.9, 7.8, and 3.6%). Among the 29 ICT-positive subjects (all of whom were re-sampled at night), only five (17.2%), living in two villages, presented blood mf. In the Bouenza department, ICT-positive individuals were found in 9 of the 13 villages, and the prevalence of antigenemia exceeded 10% in four (10.0, 13.1, 13.2, and 24.1%). Among the 129 ICT-positive individuals, 123 were re-sampled at night, and 51 (41.5%) showed mf in their blood. The highest prevalences of microfilaremia were 10.8, 5.0, and 3.7%. The village with the highest prevalence, Seke Pembe, was selected for the trial of ALB alone, conducted from 2012. In the Sangha department, only two ICT-positive subjects, living in the same village, were identified, and both showed blood mf by night. Detailed results of this study will be presented in another paper.

In 2013, the PNLO launched annual MDA combining IVM and ALB in five HDs where onchocerciasis is co-endemic with LF. These HDs were taken as implementation units (IU): Kibangou and Dolisie in the Niari department, and Madingou, Mouyondzi, and Loutete in the

Bouenza department [76]. From 2013 to 2015, the PNLO reports that ALB distribution was only partially achieved in the selected IUs.

In 2015, the PNLO conducted a new nationwide mapping of LF using the ICT as RDT and diurnal blood smears for loiasis. Briefly, two to six villages per HD were randomly selected, stratified by health area (HA), and at least 50 adults were tested in each village. An HD was considered endemic if one ICT-positive case was reported. In total, 4,879 individuals were screened in 93 villages located in 31 HDs, and very few cases were found: one case in Bos-souaka (Cuvette-Ouest), one in Okia (Cuvette), four in Botala (Likouala), and one in Yamba (Bouenza). In S8 Appendix we reported results of the survey and in S9 Appendix the number of tested individuals.

From 2016 to 2019, the PNLO administered in average 289,642 ALB treatments by MDA per year for an average TC of 79.5% (80.8% in 2016, 78.1% in 2017, 77.9% in 2018 and 81.2% in 2019) (S10 Appendix).

Following this survey, the PNLO identified eight new IUs previously not identified as endemic for LF, and where MDA for LF elimination had also to be implemented: Kimongo (Niari department), Ignié (Pool), Owando (Cuvette), Ewo and Etoumbi (Cuvette-Ouest), Sembe-Souanke and Ouesso-Mokeko-Pikounda (Sangha), and Impfondo (Likouala). In 2019, the PNLO started the ALB MDA in previously untreated IUs, which led to higher TC at the national level, as the denominator for TC already took into account those areas not yet treated.

To address the problem of co-endemicity with loiasis, seven of the eight new IUs (Kimongo, Impfondo, Ouesso, Sembe-Souanke, Owando, Ewo, and Etoumbi) have been treated with ALB alone twice a year. In the HDs of Ignié (Pool department), Madingou, Mouyondzi and Loutete (Bouenza), and Dolisie and Kibangou (Niari), either ALB alone or ALB combined with IVM have been administered, on a village-by-village basis.

Table 2 summarizes the studies on LF included in this review. In October 2012, the total population of Seke Pembe (Bouenza department) was invited to participate in a community trial to evaluate the efficacy of ALB alone (given at a single dose of 400 mg) to eliminate LF. In total, 773 individuals were tested by ICT before the first treatment, and those found CFA-positive were resampled at night for mf examination. The initial antigenemia and microfilaremia prevalence values were 17.3% and 5.3%, respectively, and results obtained during the annual follow-up examinations supported the use of semi-annual MDA with ALB alone to eliminate LF in loiasis co-endemic areas where IVM cannot be safely distributed [77, 78].

## Soil-transmitted Helminthiases

**Clinical presentation and diagnostic methods.**   Among the NTDs, STHs represent the group with the highest morbidity burden [10]. The main STHs are ascariasis (caused by *Ascaris lumbricoides*), trichuriasis (caused by *Trichuris trichiura*), hookworm infection (caused by *Ancylostoma duodenale* and *Necator americanus*), and strongyloidiasis (caused by *Strongyloides stercoralis*). The humid climate of many tropical countries, and the unsatisfactory levels of hygiene and sanitation measures, are conducive to the development of eggs and larval stages of these parasites [83]. STHs affect mainly children aged 0–15 years, but high prevalence can also be found in adults, especially for hookworms [83]. Ascariasis is often associated with abdominal distension and pain, but complications such as complete intestinal obstruction, intestinal perforation, or peritonitis can occur. Trichuriasis can cause chronic abdominal pain and diarrhea, and high infection can lead to chronic dysentery and rectal prolapse. Hookworm infection is often asymptomatic but can induce marked anemia, hypo-proteinemia, and growth retardation in children and exacerbate pre-existing anemia in pregnant women.

**Table 2. Summary of the included epidemiological studies and reports on LF.**

| Study | Year of survey | Departments | Villages | N* | Main results$ |
|---|---|---|---|---|---|
| **Therapeutic assessment surveys (WER reports)** | | | | | |
| 2017 [79] | 2017 | Niari, Bouenza, Pool | Niari (Kibangou and Dolisie) Bouenza (Madingou, Mouyondzi and Loutete) Pool (Ignié) | 136,373 | TC by PNLO: 78.1% TC reported by the WER: 77.8% |
| 2016 [80] | 2016 | Niari, Bouenza, Pool | Niari (Kibangou, Dolisie, and Kimongo) Bouenza (Madingou, Mouyondzi and Loutete) Pool (Ignié) | 551,879 | TC by PNLO: 80.8% TC reported by the WER: 20.2% |
| 2015 [81] | 2015 | Niari, Bouenza | Niari (Kibangou and Dolisie) Bouenza (Madingou, Mouyondzi and Loutete) | 126,672 | TC reported by the WER: 91.9% |
| 2014 [82] | 2014 | Niari, Bouenza | Niari (Kibangou and Dolisie) Bouenza (Madingou, Mouyondzi and Loutete) | 126,363 | TC reported by the WER: 82.4% |
| 2013 [76] | 2013 | Niari, Bouenza | Niari (Kibangou and Dolisie) Bouenza (Madingou, Mouyondzi and Loutete) | 111,756 | TC reported by the WER: 92.8% |
| **Prevalence surveys** | | | | | |
| PNLO report, 2015 | 2015 | National level | National level | | 7 cases of LF at the national level |
| Pion et al., 2017 [78] | 2012 | Bouenza | Séké-Pembé | 773 | CFA prevalence value: 17.3% mf prevalence value: 5.3% |
| Carme et al., 1986 [73] | 1981 | Brazzaville, Pool, Likouala, Kouilou, Lekoumou | Brazzaville, Pool (Mpayaka Kibouende, Mayama, Ntombo Manyanga, Linzolo), Likouala (Impfondo), Kouilou (Mvouti, Loandjili), Lekoumou (Sibiti, Zanaga, Komono) | 17,841 | No reported case of LF |

* All these surveys were cross-sectional studies and studies by Pion et al. were conducted as part of a community trial.

$ TC: therapeutic coverage; LF: lymphatic filariasis; CFA: Circulating Filarial Antigens; mf: microfilariae; WER: Weekly Epidemiological Reports.

The classical diagnosis of STH is based on microscopic examination of stool samples, and the gold standard for diagnosis is the Kato-Katz method [84]. However, several other diagnostic techniques exist, each with its advantages and disadvantages [85]. PCR-based techniques are particularly interesting because they are highly sensitive and specific, enable accurate species and strain identification, and help monitor transmission patterns through molecular epidemiology [85]. Though effective, PCR requires well-trained biologists in well-equipped laboratories, which are lacking in remote areas of developing countries.

STH treatment is based on the administration of drugs belonging to the benzimidazole family. A single dose of ALB (400 mg) is very effective for ascariasis and hookworm, but a 3-day course is required to cure trichuriasis [86]. A single dose of mebendazole (500 mg) is as effective as ALB on ascariasis, less effective on hookworms, and, again, multiple doses of the drug must be used to achieve satisfactory cure rates on trichuriasis [86].

**History of epidemiological surveys.** The first reported epidemiological study on STHs in the RoC, carried out between August 1952 and March 1953, aimed to assess the level of endemicity for various parasitic diseases in Brazzaville [87]. Analysis of samples collected from 551 children and 401 adults in the central districts of Poto-Poto, Bacongo, and Ouenzé found the prevalence of infection with *A. lumbricoides* was 44.4% (45.6% in children and 42.9% in adults) and 61.8% with hookworm was (56.4% in children and 69.1% in adults). Much lower prevalence values were recorded for infections with *T. trichiura* (7.8% overall, 8.9% in children, and 6.2% in adults) and *S. stercoralis* (5.1% overall, 3.6% in children, and 7.2% in adults). In 1984, another Brazzaville study of 230 children aged 2–14 years reported a hookworm

prevalence of 24.3%, with boys more often infected than girls (29.8 vs. 19.0%) [88]. A survey conducted on 418 children aged 1–6 years living in eastern districts of Brazzaville revealed overall prevalence values of 24%, 32%, 2%, and 4% for *A. lumbricoides*, *T. trichiura*, hookworms, and *S. stercoralis*, respectively, and demonstrated substantial differences among districts due to environmental factors [89].

Information on STH prevalence outside Brazzaville is exceptionally scarce. In 1966, Davadie et al. evaluated the prevalence of various parasitic diseases in 223 individuals in the town of Dolisie (Niari department) and 208 in Kayes (Bouenza department). The prevalence of ascariasis in these localities was 66.8 and 58.7%, respectively, that of trichuriasis was 86.1 and 94.2%, that of hookworm infection, 51.1 and 15.4%, and that of strongyloidiasis, 8.9 and 1.4% [90]. The only other data retrieved are those collected in 1988 from laboratories in 7 localities: Ouesso and Souanké in the Sangha department, Impfondo, Epena and Dongou in the Likouala department, and Owando and Mossaka in the Cuvette department. Prevalence values ranged between 37.5 and 76.3% for *A. lumbricoides*, 5.7 to 28.0% for *T. trichiura*, 1.7 to 37.5% for hookworms, and 0 to 3.9% for *S. stercoralis* [91].

**Epidemiological surveys and MDAs organized by the PNLO.** No further information on STHs was available until 2011, when the PNLO conducted a nationwide mapping of STH and schistosomiasis prevalence. In total, 18,894 children (5–15 years) from 324 schools were examined. These schools were located in 103 districts and arrondissements in the 12 departments of the country. The parasite species of eggs found in stool samples were not distinguished, and thus, the prevalence measured was that of "at least one STH." Prevalence exceeded 20% in all departments except the Brazzaville, Cuvette, and Pointe-Noire, where the prevalences were 6.6, 17.7, and 14.4%, respectively. High prevalence values exceeding 70% were observed in thirteen AD had including Sembe (81.8%) and Souanke (84.5%) in the Sangha department; Djambala (72.2%), Lekana (78.0%), and Mpouya (71.8%) in the Plateaux department; Komono (75.0%) and Bambama (84.7%) in the Lekoumou department; and Moutamba (73.3%), Nyanga (78.4%), Banda (84.7%), Divenie (79.8%), Londelakayes (92.5%), and Kimongo (79.0%) in the Niari department (S11 Appendix). A map summarizing the results is presented in Fig 7.

A 2012 survey of the entire population of the village of Seke-Pembe (see above) found the prevalence of hookworm, *A. lumbricoides*, and *T. trichiura* infections to be 6.5, 56.4, and 78.6%, respectively. Subsequent biannual population-wide mass treatment with ALB (400 mg) led to a significant reduction in hookworm infections one year after the first distribution [77]. At the end of the third year of treatment, both hookworm and *T. trichiura* infections had decreased [78].

Between 2014 and 2018, the PNLO organized the distribution of ALB (400 mg) to 1,913,089 school-aged children (SAC) as preventive chemotherapy against STHs. This came to an average of 382,618 per year and an average TC of 80.7% (S12 Appendix). The only year when coverage dipped below the recommended 75% was 2015, when TC was 72.3%. Table 3 summarizes the studies on STHs retrieved for this review.

## Schistosomiasis

**Clinical presentation and diagnostic methods.** Schistosomiasis is a parasitic disease transmitted to humans through contact with water containing infective stages (cercariae) of *Schistosoma* worms released by gastropods. Six species of schistosomes are pathogenic to humans: *S. haematobium*, inducing urogenital manifestations, and five others causing intestinal or hepatic manifestations: *S. mansoni*, *S. japonicum*, *S. intercalatum*, *S. mekongi*, and *S. guineensis*. Depending on the schistosome species involved, the disease can lead to

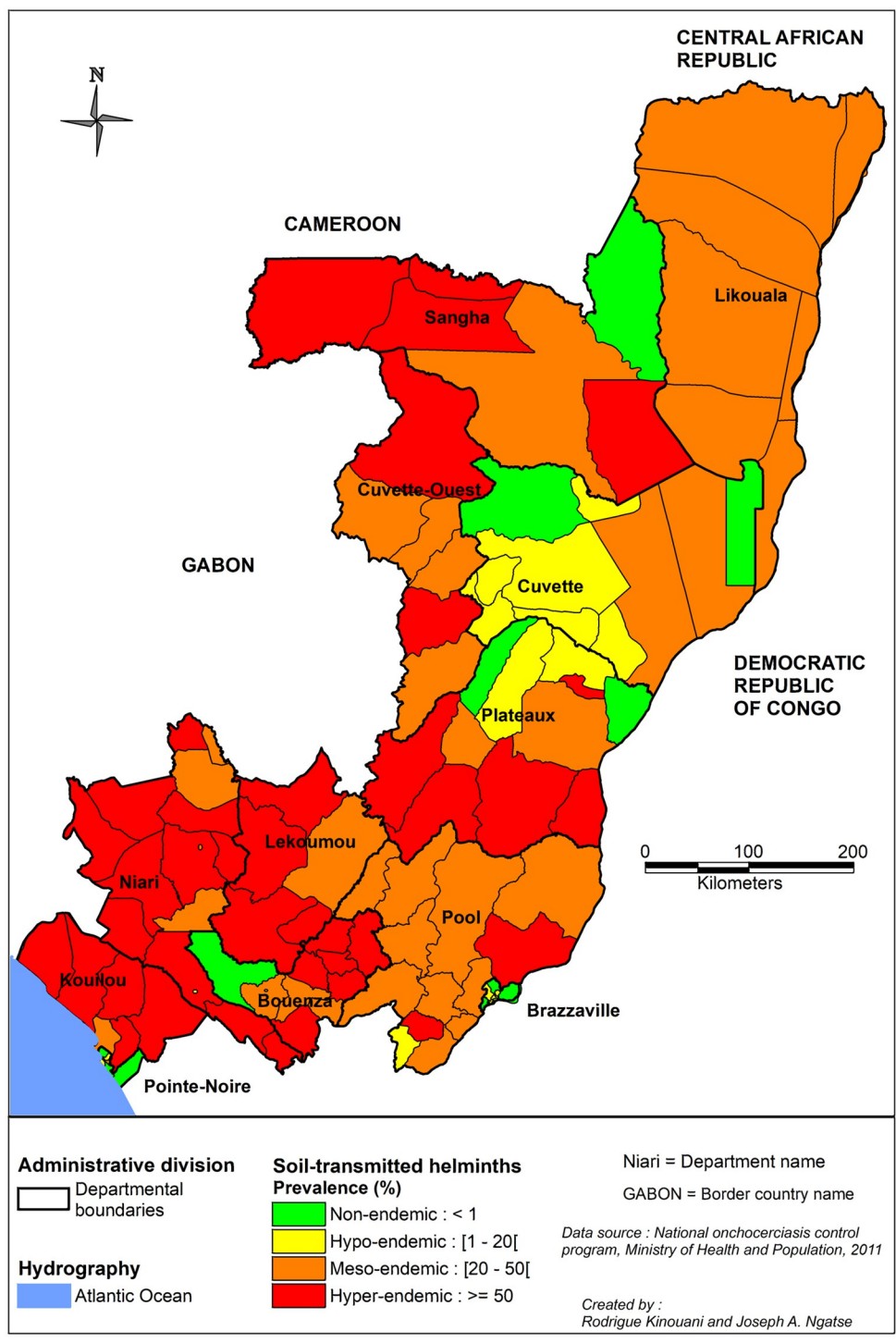

**Fig 7. Distribution of soil-transmitted helminths infections in the Republic of the Congo.** Data were obtained from the 2011 national survey performed by the PNLO, which used the Kato-Katz method for diagnosis. Reported prevalence values are for at least one STH. The map was created with MapInfo 8.5 (Geographic Information System, http://www.precisely.com). The base layer used of the map was created by the « Laboratoire Population Environnement Développement » (LPD, UMR 151 AMU-IRD) (https://www.lped.fr/-observatoires-societe-environnement-.html) under the supervision of the Ministry of Health and Welfare of the Republic of Congo.

**Table 3. Summary of the included epidemiological studies and reports for STH.**

| Study | Year of survey | Departments | Villages | N | Main results[$] |
|---|---|---|---|---|---|
| Pion et al., 2017 [78] | 2012–2015 | Bouenza | Seke-Pembe | 350 | Significant decrease in the arithmetic mean number of eggs per gram of stool between 2012 and 2015:<br>• reduction by 66.9% for *T. trichiura*<br>• reduction by 92.6% for *A. lumbricoides*<br>• reduction by 100% for hookworms |
| Pion et al., 2015 [77] | 2012–2013 | Bouenza | Seke-Pembe | 335 | Significant decrease in prevalence values between 2012 and 2013:<br>• reduction by 35.5% for *A. lumbricoides*<br>• reduction by 9.8% for *T. trichiura*<br>• reduction by 90.7% for hookworms |
| PNLO, 2011 | 2011 | National level | National level | 4,222,164 | See Map in Fig 5 |
| Dorier-Apprill, 1993 [89] | 1993 | Brazzaville | Brazzaville | 5336 | Ascariasis: 24%; Trichuriasis: 32%; Hookworm infection: 2%; Strongyloidiasis: 4% |
| Mengho B, 1988 [91] | 1988 | Sangha, Likouala, Cuvette | Ouesso and Souanke (Sangha), Impfondo, Epena and Dongou (Likouala), Owando and Mossaka (Cuvette) | 11,742<br>Ouesso: 5,020<br>Souanke: 1,324<br>Impfondo: 1,354<br>Epena: 948<br>Dongou: 1,137<br>Owando: 619<br>Mossaka: 1,340 | 1-**Ascariasis**: Ouesso (41.6%), Souanke (64.2%), Impfondo (37.5%), Epena (76.3%), Dongou (55.9%), Owando (61.5%), Mossaka (53.7%)<br>2-**Hookworm** infection: Ouesso (28.7%), Souanke (1.7%), Impfondo (37.5%), Epena (12.0%), Dongou (31.9%), Owando (25.5%), Mossaka (17.2%)<br>3-**Trichuriasis**: Ouesso (27.7%), Souanke (28.0%), Impfondo (19.3%), Epena (11.7%), Dongou (11.7%), Owando (5.7%), Mossaka (11.3%)<br>4-**Strongyloidiasis**: Ouesso (2.0%), Souanke (1.1%), Impfondo (1.1%), Owando (1.6%), Mossaka (3.9%) |
| Carme, 1984 [88] | *Not specified* | Brazzaville | Brazzaville | 230 | Global prevalence: 24.3% |
| Davadie et al., 1966 [90] | 1966 | Bouenza and Niari | Kayes, Dolisie, Jacob and Loudima-Gare | Kayes: 480<br>Dolisie: 223 | Ascariasis: Dolisie (66.8%) and Kayes (58.7%)<br>Trichuriasis: Dolisie (86.1%) and Kayes (94.2%)<br>Hookworm: Dolisie (51.1%) and Kayes (15.4%)<br>Strongyloidiasis: Dolisie (8.9%) and Kayes (1.4%) |
| Lamy et al., 1954 [87] | 1952–1953 | Brazzaville | Brazzaville | 1511 | Ascariasis: 44.0%<br>Hookworm infection: 61.7%<br>Few infections with *T. trichiura* |

[$] Percentages correspond to the prevalence rates; all these surveys were cross-sectional studies and studies by Pion et al. were conducted as part of a community trial.

hepato-splenomegaly, hematuria, bladder cancer, or even sterility, and facilitate infection with HIV [92].

The classical diagnosis is based on the detection of eggs in urine or feces by microscopy. More recently, tests detecting parasite-secreted circulating anodic or cathodic antigens (CAA and CCA) in serum and urine have been developed. These tests are highly specific and sensitive, and a point-of-care CCA urine cassette test for detecting intestinal schistosomiasis is commercially available. Other tests detecting parasite DNA in urine or feces by polymerase chain reaction (PCR) or loop-mediated isothermal amplification (LAMP) technologies have also been developed [93].

The recommended treatment is a single dose of praziquantel (40 mg/kg for *S. haematobium* and *S. mansoni*, and 60 mg/kg for *S. japonicum* and *S. mekongi*) [93]. MDAs of praziquantel target principally SAC and adults exposed to a risk of infection, and the interval between MDA depends on the initial prevalence of infection.

Schistosomiasis endemicity is defined at the district level and determined by estimating the prevalence of infection in SAC in five selected schools. Schistosomiasis is considered

non-endemic when the mean prevalence is <1%, and low, moderate, and high risk when the mean prevalence values are 1–9.9%, 10–49.9%, and ≥50%, respectively [94]. MDA is not organized in low-risk areas, but praziquantel is made available in the health structures to treat suspected cases. In districts with moderate or high risk, a 2013 WHO progress report [95] recommended praziquantel MDA once every two years or once a year, respectively. In a more recent guideline, issued in 2022 [96], WHO recommends praziquantel MDA once a year or twice a year for moderate and high risk districts, respectively.

**History of epidemiological surveys.** It was suggested that schistosomiasis was introduced in the RoC in the 1920s by foreign workers, particularly Senegalese and Chadian, recruited for the 1921 to 1934 construction of the "Congo-Ocean" railroad between Brazzaville and Pointe-Noire. This hypothesis is based on the 1953 observations in Brazzaville and its surrounding areas of Lamy, who reported that infections with *Schistosoma* sp. were not observed in Brazzaville natives, but were observed among the foreign workers. The prevalence of *S. mansoni* infection was 18.9% among workers from the CAR and 2.5% for those from Chad. The prevalence of *S. haematobium* infection in these two subpopulations was 4.3 and 50.8%, respectively [97].

The first survey on schistosomiasis in the RoC was conducted in 1920 in about 30 villages located on the right bank of the Ubangui and Congo rivers, between Bangui (CAR) and Loukolela, thus mostly in the present Likouala department. The author collected 500 stool and 400 urine samples and estimated he had examined about 10% of the population in the surveyed villages. Schistosomiasis was diagnosed only in the small village of Irebou, with a population of about 50 people, located near the border separating the Likouala and the Cuvette departments. "*S. haematobium*" eggs were observed in the feces of five people from the village, but no egg in the urine. It was hypothesized that these infections may have originated on the other side of the Congo River, in the Democratic Republic of Congo (formerly Zaire), where the people regularly traveled [98]. The absence of eggs in the urine and information from later sampling in the same area suggest that the parasite present in this focus (around Impfondo, Irebou, and Loukolela) was *S. intercalatum*, not *S. haematobium* [99, 100].

In 1964, McCullough reviewed data collected between 1956 and 1962 on schistosomiasis in the RoC. Hospital data and results of surveys conducted on SAC by the Ministry of Health suggest small endemic foci of *S. haematobium* in the Niari, Bouenza, and Kouilou departments, particularly around the towns of Nkayi (formerly Jacob), Loudima, Dolisie, and Kibangou. The author could not confirm the endemicity of intestinal schistosomiasis (due to *S. mansoni*) in the RoC, despite a few reported cases from the Prefecture of Djoué (i.e., in the Brazzaville area) [101]. A 1966 survey of Nkayi and Kayes, small villages located 2 km apart on the bank of the Niari river, and Loudima and Dolisie reported no infections with *S. mansoni*. However, the prevalence of *S. haematobium* infection ranged between 90% to 99% in these villages, with 76% of the subjects aged < 20 and 40% of those aged ≥ 20 years infected. The authors highlight the marked increase in the prevalence values of *S. haematobium* infection in the Bouenza-Niari focus compared to previous data [90]. These results were confirmed by a retrospective study of the cases diagnosed and reported in the "*Grandes Endémies*" units between 1963 and 1976 (Fig 8) [102]. This study showed that *S. haematobium* affected mainly the Niari and Bouenza departments (prevalence values > 30%), whereas *S. mansoni* mainly affected Brazzaville (prevalence > 50%), as well as the Kouilou department (prevalence > 10%) [102]. These high prevalence values led to the creation of the National schistosomiasis control program [103].

Fig 9 shows historical and recent prevalence rates of urinary schistosomiasis in the endemic departments of the RoC. Historical data is from the 1987 review by Doumenge et al. [103]. The recent data is from the last nationwide surveys conducted in 2011 on SAC aged 5–15 years (see below).

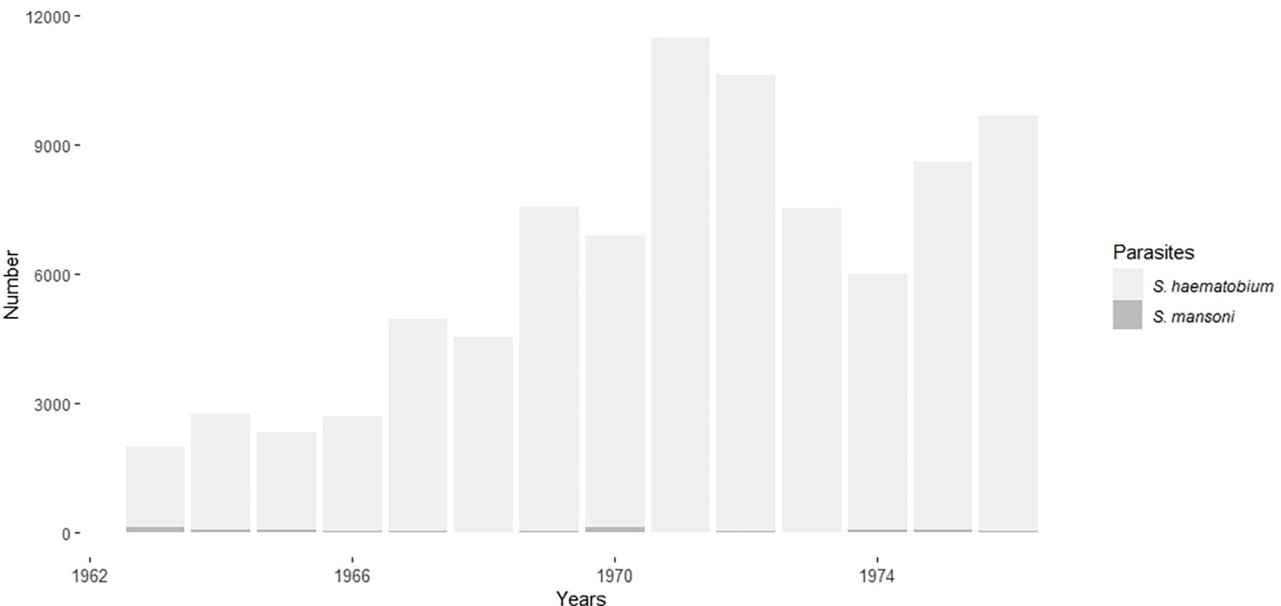

**Fig 8. Distribution of schistosomiasis cases reported over the years.** The number of urogenital and intestinal schistosomiasis cases reported in a retrospective epidemiological study conducted between 1963 and 1976 [102].

After the creation of the schistosomiasis control program in 1986, additional surveys were conducted in the RoC. In Brazzaville, a focus of urinary schistosomiasis was reported in 1987 in a quarter called "Plateau des 15 ans", near the Mfoa River, where 8.2% of the 5733 school-children examined were infected (the prevalence values were 16.4 and 2.3% in boys and girls, respectively) [104]. This focus was confirmed by a second survey [105]. In 1986–1987 a survey was conducted in villages in two regions of the Kouilou department. First, in the Mayombe forest area, high prevalence (20%) was observed in only one (Les Saras) of five villages, among school children (aged 6–20 years). By contrast, high prevalence was reported in four of the six villages surveyed located near lakes, with prevalence values ranging between 20% and 66% for the total population [106].

**Epidemiological surveys and MDAs organized by the PNLO.** In 2011, the nation-wide schistosomiasis (and STH) mapping survey conducted on SAC reported prevalence values higher than 20% in most of the districts of the Kouilou department (including Madingo-Kayes with a prevalence of 58.8%) as well as in Mouyondzi and Nkayi districts in the Bouenza department (S11 Appendix). The geographic distribution of schistosomiasis according to this data is shown in Fig 9 (right panel). It should be noted that the prevalence values were higher among children than adults.

Table 4 summarizes the studies on schistosomiasis included in the present review. In 2014, annual MDA with praziquantel targeting the SAC (5–14 years) was launched in endemic HDs. In 2015, this distribution covered the entire department of Kouilou, the Sibiti HD in the Lekoumou department, and three HDs in the Bouenza department: Nkayi, Loudima, and Mouyoudzi. In 2016, MDA was restricted to the Kouilou department. From 2014 to 2018, the PNLO administered an average of 28,961 treatments per year, for an average TC of 73.3%. The lowest coverage rates (and the only ones under the recommended level of 75%) were recorded in 2015 (54.6%) and 2017 (64.4%). Overall, between 2014 and 2018, the PNLO administered

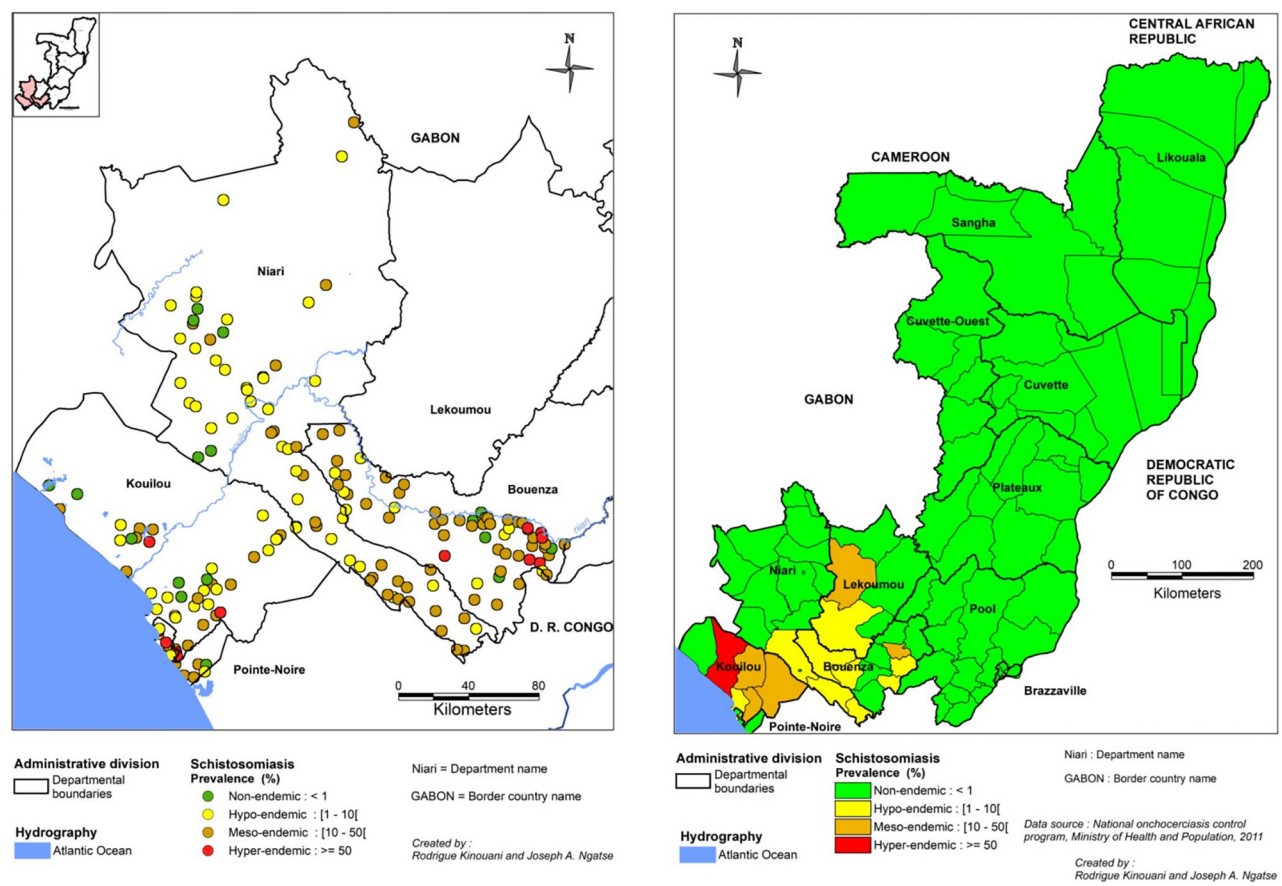

**Fig 9. Distribution of schistosomiasis in the Republic of the Congo.** The left panel reports results from all epidemiological surveys conducted between inception to 1987 (last known published study); the right panel reports the most recent results, from a 2011 survey conducted by the PNLO. The map was created with MapInfo 8.5 (Geographic Information System, http://www.precisely.com). The base layer used of the map was created by the « Laboratoire Population Environnement Développement » (LPD, UMR 151 AMU-IRD) (https://www.lped.fr/-observatoires-societe-environnement-. html) under the supervision of the Ministry of Health and Welfare of the Republic of Congo.

144,804 praziquantel treatments (S13 Appendix). In 2019, no MDA was organized due to lack of funding, and in 2020 the PNLO administered 62,350 praziquantel treatments for an average TC of 69.1%. There has not yet been a distribution of treatment for 2021.

## Trachoma

**Clinical presentation and diagnostic methods.** Trachoma is an ocular infection caused by *Chlamydia trachomatis*, transmitted from person to person through contaminated hands or clothes and by flies that were in contact with the runny nose or eyes of an infected person. It occurs and is always endemic in remote areas and poorest populations of Africa, Asia, Australia and the Middle East resulting in blindness [110]. Active trachoma mainly affects children under 5 years of age and its prevalence decreases with age [110]. In the absence of care following repeated infections, scars appear on the eyelid, leading to a distortion of the inner part of the eyelid and the contact of the eyelashes with the cornea (trichiasis). Trichiasis is painful and, if left untreated, can lead to blindness (trachoma is the leading infectious cause of blindness worldwide) [110]. The London Declaration of 30[th] January 2012 called for the eradication of blinding trachoma by 2020 [111] and the WHO now targets its elimination as a public health problem by 2030 [1].

**Table 4. Summary of the epidemiological studies and reports on schistosomiasis included in this review.**

| Study | Year of survey | Departments | Villages | Design | N | Main results$ |
|---|---|---|---|---|---|---|
| **Reports** | | | | | | |
| PNLO, 2011 | 2011 | National level | National level | Cross-sectional | 4,222,164 | See Map Fig 9 |
| WHO, 1995 [107] | 1990–1991 | Bouenza | Nkayi | Cross-sectional | 5590 school-children | Hematuria: 18.3% |
| **Prevalence surveys** | | | | | | |
| N'zoukoudi-N'doundou et al., 1994 [108] | 1992 | Bouenza | Nkayi | Cross-sectional (relationship between HIV and schistosomiasis) | 895 | Urinary schistosomiasis: 38.1% [34.9–41.3] |
| Akouala et al., 1989 [106] | (a) 1986 (Mayombe forest area); (b) 1987 (villages near lakes) | Kouilou et Pointe-Noire | (a) Mvouti, Les Saras, Dimonika, Makaba, Mpounga (b) Mboukou-Massi, Mbouyou, Wolo, Nanga-Lac, Nanga-Mpili, Kayo | Cross-sectional | Mayombe (n = 686) Littoral (n = 1797) | Urinary schistosomiasis: (a) only one village (Les Saras) had a fairly high prevalence (20.0% [95%CI: 12.8–27.2]) (b) four had a prevalence higher than 20%: Wolo (20.5%), Mbouyou (42.5%), Nanga-Lac (65.5%), and Mboukou-Massi (65.5%) |
| Akouala et al., 1988c [104] | 1988 | Brazzaville | Plateaux /Loutassi | Cross-sectional | 5733 | Urinary schistosomiasis: 8.2% [7.5–8.9]; Most cases were contaminated in Brazzaville (90.6%) |
| Akouala et al., 1988b [105] | 1987–1988 | Brazzaville | Plateaux | Cross-sectional | 1006 from Plateaux (school 1); 434 from Mboueta-Mbongo school (school 2) | Urinary schistosomiasis: school 1: 11.1% [9.2–13.1] and school 2: 10.6% [7.7–13.5]; 69.6% and 93.5% of the positive cases from schools 1 and 2, respectively, had never left Brazzaville. |
| Akouala et al., 1988a [109] | 1987 | Pool | Loulombo (formerly De Chavannes) | Cross-sectional | 1337 | Urinary schistosomiasis: 15.6% [13.7–17.6]; 87.5% of the positive cases had never left Loulombo. |
| Doumenge et al., 1987 [103] | Inception to 1987 | Bouenza, Niari, Kouilou | Bouenza, Niari, Kouilou | Retrospective review | | See Map Fig 7 |
| Ngaporo and Coulm, 1978 [102] | 1952–1976 | National level | National level | Retrospective study | 95,434 cases infected with *S. haematobium*, 910 cases infected with *S. mansoni* | **Number of cases infected with *S. haematobium*** **1963**: 1866 (Ni 49%, Bo 40%, Bz 2%) **1964**: 2695 (Ni 41%, Bo 38%, Po 13%) **1965**: 2267 (Ni 56%, Bo 32%, Le 5%) **1966**: 2661 (Bo 47%, Ni 39%, Po 7%) **1967**: 4935 (Bo 54%, Ni 30%, Po 5%) **1968**: 4512 (Ni 39%, Bo 34%, Ko 14%) **1969**: 7496 (Bo 73%, Ni 20%, Ko 11%) **1970**: 6756 (Bo 66%, Bz 49%, Ni 16%) **1971**: 11466 (Bo 74%, Ni 13%, Ko 7%) **1972**: 10594 (Bo 61%, Ni 23%, Ko 8%) **1973**: 7523 (Bo 53%, Ni 29%, Ko 11%) **1974**: 5916 (Bo 63%, Ni 20%, Ko 10%) **1975**: 8534 (Ni 50%, Bo 33%, Ko 9%) **1976**: 9640 (Bo 63%, Ko 18%, Ni 13%) **Number of cases infected with *S. mansoni*** **1963**: 113 (Ni 80%, Bz 13%, Ko/Pl 2%), **1964**: 57 (Bz 74%, Pl 14%, Ko/Li 5%) **1965**: 69 (Le 45%, Ni 33%, Po 11%) **1966**: 51 (Bz 69%, Ko 16%, Ni 12%) **1967**: 37 (Bo 43%, Bz 24%, Li 8%) **1968**: 19 (Bz 74%, Le 16%, Cu 10%) **1969**: 49 (Bz 92%, Ko 4%, Bo/Pl 2%), **1970**: 138 (Ko 80%, Bz 19%, Bo 1%) **1971**: 20 (Bz 55%, Ko 40%, Le 5%) **1972**: 38 (Ni 37%, Ko 34%, Bz 29%) **1973**: 9 (Bz 89%, Ko 11%) **1974**: 74 (Bz 81%, Ko 16%, Bo/Ni 1%) **1975**: 64 (Bz 81%, Cu 9%, Ko/Ni 5%) **1976**: 30 (Bz 60%, Po 30%, Ni 7%) |
| Davadie et al., 1966 [90] | 1966 | Bouenza and Niari | Kayes, Dolisie, Jacob and Loudima-Gare | Cross-sectional | Kayes: 480 Dolisie: 223 Nkayi (formerly Jacob): 60 Loudima-Gare: 100 | Urinary Schistosomiasis: Prevalence in Dolisie: 22% Prevalence in Kayes: 41.7% Prevalence in Nkayi: 41.6% Prevalence in Loudima-Gare: 10% |

*(Continued)*

**Table 4.** (Continued)

| Study | Year of survey | Departments | Villages | Design | N | Main results$ |
|---|---|---|---|---|---|---|
| McCullough, 1964 [101] | 1955–1961 | National level | National level | Retrospective study | | **Number of cases infected with *S. haematobium*** <br>**1956**: 795 (Ni 81%) <br>**1957**: 1276 (Ni 85%, Ko 13%) <br>**1958**: 907 (Ni 74%, Ko 19%) <br>**1959**: 992 (Ni 61%, Ko 27%) <br>**1960**: 1339 (Ni 71%, Ko 15%) <br>**1961**: 1632 (Ni 83%) <br>**Number of cases infected with *S. mansoni*** <br>**1956**: 2 (Ko 100%) <br>**1957**: 13 (Ni-Bo 54%) <br>**1958**: 9 (Ni-Bo 33%) <br>**1959**: 13 (Dj 46%, Li 31%) <br>**1960**: 13 (Dj 46%, Li-Mo 23%) <br>**1961**: 9 (Dj 100%) |
| Lamy, 1953 [97] | 1952–1953 | Brazzaville | Brazzaville | Cross-sectional | 2000 | No infection with *S. mansoni* nor *S. haematobium* among the Congolese nationals; *S. haematobium* in Nationals of Oubangui-Chari (CAR): 4.3% (2/46) and Nationals of Chad: 50.8% (64/126); and *S. mansoni* in Nationals of Oubangui-Chari (CAR): 18.9% (14/74) and Nationals of Chad: 2.5% (2/80) |

$ Prevalence (%) and [95% CI]**; Bo**: Bouenza; **Bz**: Brazzaville; **Dj**: Djoué; **Li-Mo**: Likouala-Mossaka; **Ni**: Niari; **Ni-Bo**: Niari-Bouenza; **Pl**: Plateaux; **Po**: Pool

Although sensitive and specific nucleic acid amplification techniques are available, no diagnostic test for trachoma is used as a reference, and diagnosis remains clinical. Treatment is based on a single annual dose of azithromycin (20 mg/kg to a maximum of 1 g for adults) [110]. Treatment of infants under 6 months of age is classically based on topical application of tetracycline ointment daily for 6 weeks [112, 113].

**History of epidemiological surveys.** In 2013, trachoma was not considered endemic in the RoC [114], despite the Likouala department bordering areas of high trachoma prevalence in the CAR [115]. The only trachoma survey in the RoC, conducted in 2015 by the PNLO, aimed at assessing the prevalence of trachomatous inflammation-follicular (TF) in children aged 1–9 years and trachomatous trichiasis (TT) in adults aged ≥15 years in two ADs of the Sangha department and five ADs of the Likouala department [116]. When considering these seven districts as a single assessment unit, the prevalence of TF was 2.5%, and no case of TT was reported. These results, presented in table 5, confirmed that the disease is not a public health problem in this part of the RoC, according to WHO criteria (TF < 5% and TT < 0.2%) [113]. However, prevalence of TF exceeded 5% in some villages (see S7 Appendix). Systematic surveys should be conducted at the national level to properly assess the burden of the disease in the country.

**Table 5. Summary of the epidemiological studies and reports for trachoma.**

| Study | Year of survey | Departments and villages | Design | N | Main results$ |
|---|---|---|---|---|---|
| Missamou et al., 2018 [116] | 2015 | Likouala (Betou, Dongou, Enyelle, Impfondo, Epena) and Sangha (Ouesso, Mokeko) | Cross-sectional | 1222 between 1 and 9 years and 694 adults | Trachomatous inflammation-follicular (2.5 [0.9–4.5]) and trachomatous trichiasis (0) |

$ Prevalence (%) and [95% CI]

## Discussion

The present historical review is intended to document the past and current situation of PC-NTDs in the RoC and identify challenges to eliminating NTDs. It shows that some NTDs (onchocerciasis, STH, and schistosomiasis) have been reported in the RoC for several decades, whereas others (LF, trachoma) were investigated and found to be present only more recently. As expected, prevalence values varied widely across the country, depending on various ecological factors (e.g., vegetation or hydrography) and the lifestyle of the communities concerned.

Over the years of control activities, the status of these diseases has dramatically changed. The REMO survey conducted in 2001 to assess the national PNod revealed hyper-endemicity of onchocerciasis in the southern part of the RoC, particularly in the districts of Boko and Kinkala (Pool Department) and Divenie (Niari Department). The annual IVM MDA that followed this evaluation resulted in a clear decrease in PNod in these endemic departments, as reported between 2011 and 2015, despite some persistent hyper- and meso-endemic foci. These observations call for a new evaluation of the extent of onchocerciasis endemicity, after almost 20 years of CDTI, in areas where large-scale interventions were not done (such as in the Kouilou department).

In addition, to eliminate onchocerciasis, as targeted in the WHO's latest roadmap for NTDs (2021–2030) [117], the RoC needs to adopt alternative strategies for safely treating populations living in onchocerciasis hypo-endemic areas where loiasis is co-endemic. Using a "Loa-first Test-and-Not-Treat" strategy would enable safe onchocerciasis treatment with ivermectin by identifying and excluding those few individuals harboring very high *L. loa* microfilaremia in these populations [118]. Alternatively, an "Oncho-first" test-and-treat strategy, which consists in identifying and treating (with ivermectin and/or the macrofilaricidal drug doxycycline) those few individuals infected with *O. volvulus*, could be applied [49]. This would improve national TC. The RoC may be positioned to assess and compare new useful alternative strategies to resolve the most effective treatment strategy in hypo-endemic areas.

Regarding LF, the demonstrated efficacy of biannual ALB treatment in areas co-endemic with loiasis led the PNLO to adopt this regimen in all areas where onchocerciasis is non-endemic. The most recent 2015 mapping of LF distribution showed very low numbers of antigenemia-positive individuals at national level, with no hotspots in any district. However, given the beneficial effect on STHs, ALB treatment should continue even in the case of low-level LF endemicity. Indeed, STHs, which are supposed to be eliminated as a public health problem, remain endemic at the departmental level, except for Brazzaville and Pointe-Noire, the country's two main cities, and several AD in the Plateaux, Cuvette, Bouenza, and Likouala departments. This situation calls for a stronger political commitment in the fight against STHs.

The situation for schistosomiasis changed little between the 1987 and 2011 surveys. This persistence is mainly due to the schistosomiasis control program having long remained unmanaged. Praziquantel MDA among school children, conducted by the PNLO, did not resume on a large scale until 2016. Despite there being areas where trachoma was known to be endemic, the conclusion following an evaluation of trachoma prevalence in Sangha was that the disease was not endemic at the department level. This observation should lead public authorities and international organizations to consider combatting this NTD at the local level (HA, HD) and not at the departmental or national level. Additional trachoma rapid assessment surveys should also be conducted in other departments where the disease could be present. The fight against trachoma should emphasize access to high-quality surgery to manage trachomatous trichiasis cases, as reported by the WHO in its 2021–2030 roadmap [117].

Previous literature reviews on the distribution of NTDs have been conducted in other African countries such as South Sudan, Mozambique, and Ethiopia [119–121]. PC-NTDs remain

serious public health problems in these three countries, and their distribution similarly depends on ecological and demographic factors. This is especially the case in Mozambique, where LF, STH, and schistosomiasis are found in northern and central provinces with the highest population densities and poverty indices. In Ethiopia, MDA against STHs started in 2004 and reached more than 11 million people by 2009, and for onchocerciasis, since starting MDA in 2001, TC has generally remained above the WHO's 80% threshold. In addition, Ethiopia has successfully eliminated trachoma from some areas using the WHO-recommended SAFE (surgery, antibiotics, face and environmental hygiene) strategy. Still, more action is needed in this country for LF treatment and schistosomiasis mapping. In South Sudan, it is reported that data from health facilities cannot accurately reflect the distribution of NTDs.

As a limitation of this review, we have encountered difficulties in obtaining some older articles and, despite our efforts, we may have missed some unpublished documents and reports. In the review, we mentioned how researchers could clarify the distribution of some NTDs and help understand factors underlying epidemiological patterns. This was particularly the case for LF, for which assessments made before the launch of a research project were inaccurate and benefitted from preliminary surveys aimed at identifying a study site for a community trial.

Because of the limited technical infrastructure for NTD diagnosis in rural health facilities, developing sensitive and specific RDTs for each disease is essential. While the diagnostic performance of some RDTs is good [122, 123], improvements are still needed. Indeed, recent studies found that high *L. loa* microfilaremia can lead to false-positive results for the RDTs used for LF [124–126] and skin snips for onchocerciasis diagnosis [127, 128]. Current mapping of onchocerciasis and LF in central Africa should be interpreted with caution in *L. loa* co-endemic areas, and further mapping should jointly assess *L. loa* microfilarial density during these surveys. Alternatively, Ov16 RDT could be the key to fix the false positivity of skin snip related to loiasis for mapping of onchocerciasis [129]. Even if loiasis is not currently included in the WHO's list of NTDs, reports suggest that it can induce spontaneous encephalopathy [130] and a study in Cameroon suggested that it can induce excess mortality [131]. As loiasis is highly endemic in the RoC, morbidity and mortality related to this disease might be worth investigating in this country. Lastly, programs should involve more trained personnel in the sustainable control of PC-NTDs.

## Supporting information

**S1 Appendix. List of abbreviations.**
(DOCX)

**S2 Appendix. PRISMA-Checklist-review.**
(DOC)

**S3 Appendix. Mialebama, 1985 (Ref 35).**
(PDF)

**S4 Appendix. Zoomed-in REMO survey for onchocerciasis.**
(PDF)

**S5 Appendix. Targeted pop size for onchocerciasis MDA.**
(DOCX)

**S6 Appendix. Mass Drug Administration for onchocerciasis in the RoC.**
(DOCX)

**S7 Appendix. Plan directeur de lutte contre les maladies tropicales négligées (MTN) 2018–2022.**
(PDF)

**S8 Appendix. Results of the joint mapping of LF and loiasis in the RoC, 2015.**
(DOCX)

**S9 Appendix. Number of individuals tested for LF in 2015.**
(DOCX)

**S10 Appendix. Mass Drug Administration for LF in the RoC.**
(DOCX)

**S11 Appendix. Results of the joint mapping of STH and schistosomiasis in the RoC, 2015.**
(DOCX)

**S12 Appendix. Mass Drug Administration for STH in the RoC.**
(DOCX)

**S13 Appendix. Mass Drug Administration for schistosomiasis in the RoC.**
(DOCX)

## Acknowledgments

We would like to thank the directors of the control programs, as well as their collaborators, for having made some reports available and for detailed explanation of the results observed over the years. For the search of very old publications, we thank the staff of the documentation center of the Institut de Recherche pour le Développement (IRD, French National Research Institute for Sustainable Development in Montpellier).

## Author Contributions

**Conceptualization:** Cédric B. Chesnais.

**Data curation:** Joseph A. Ngatse, François Missamou, Marlhand Hemilembolo, Sébastien D. Pion, Kirsten A. Bork, Michel Boussinesq, Cédric B. Chesnais.

**Formal analysis:** Joseph A. Ngatse.

**Funding acquisition:** Joseph A. Ngatse.

**Investigation:** Joseph A. Ngatse.

**Supervision:** Gilbert Ndziessi, François Missamou, Kirsten A. Bork, Ange A. Abena, Michel Boussinesq, Cédric B. Chesnais.

**Validation:** Joseph A. Ngatse, Gilbert Ndziessi, François Missamou, Rodrigue Kinouani, Marlhand Hemilembolo, Sébastien D. Pion, Kirsten A. Bork, Ange A. Abena, Michel Boussinesq, Cédric B. Chesnais.

**Visualization:** Joseph A. Ngatse, Rodrigue Kinouani.

**Writing – original draft:** Joseph A. Ngatse.

**Writing – review & editing:** Joseph A. Ngatse, Gilbert Ndziessi, François Missamou, Rodrigue Kinouani, Marlhand Hemilembolo, Sébastien D. Pion, Kirsten A. Bork, Ange A. Abena, Michel Boussinesq, Cédric B. Chesnais.

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
