## [Decision Letter · Decision Letter 0]

11 Nov 2021

Dear Dr Chesnais,

Thank you very much for submitting your manuscript "Historical overview and geographical distribution of Neglected Tropical Diseases amenable to preventive chemotherapy in the Republic of the Congo: a systematic review" for consideration at PLOS Neglected Tropical Diseases. As with all papers reviewed by the journal, your manuscript was reviewed by members of the editorial board and by several independent reviewers. The reviewers appreciated the attention to an important topic. Based on the reviews, we are likely to accept this manuscript for publication, providing that you modify the manuscript according to the review recommendations. 

Sincerely,

Amadou Garba

Associate Editor

Justin Remais

Deputy Editor

Reviewer's Responses to Questions

**Key Review Criteria Required for Acceptance?**

**Methods**

-Are the objectives of the study clearly articulated with a clear testable hypothesis stated?

-Is the study design appropriate to address the stated objectives?

-Is the population clearly described and appropriate for the hypothesis being tested?

-Is the sample size sufficient to ensure adequate power to address the hypothesis being tested?

-Were correct statistical analysis used to support conclusions?

-Are there concerns about ethical or regulatory requirements being met?

Reviewer #1: The paper is a systematic review that has the appropriate methodology and design. The exclusion criteria are laid out clearly. However, the inclusion criteria needed to be more detailed. The identification of records, screening were well described and illustrated in figure 1. The final number of records reviews is less than 100 but has yielded substantial power and data that has been properly processed and analyzed. There were no ethical issues associated with the systematic review. The structure of the paper should be revised to insert the disease descriptions together at the beginning -mainly as a sub-section of the introduction.

Reviewer #2: This is a well researched review article with clear objectives in the introduction whaich have been achieved by the authors

**Results**

-Does the analysis presented match the analysis plan?

-Are the results clearly and completely presented?

-Are the figures (Tables, Images) of sufficient quality for clarity?

Reviewer #1: The results need to be restructured to strictly emphasize the endemicity of the PC-NTDs and provide a good understanding of each disease's extent. The results section should focus more on actual disease prevalence data and trends rather than the availability of publications and the review processes. There are significant data on treatments that are unnecessary. The paper should be focused more on the geographical distribution and historical data - as stated in the title. 

Some tables are poorly presented, many are of poor quality. Most tables should be reformatted because the characters are small and difficult to read. They are long tables and are spread over multiple pages example, tables 2,5, 6, and 7.

The onchocerciasis section would have been more helpful to present the data using nodules prevalence, CMFL, and skin snip microfilaremia.

Reviewer #2: There is a wealth of material that has been researched from the literature. This has been clearly presented in detail both in descriptive sections and particularly the tables. Figures 2 and 8 could be presented with a little more contrast to make it easier to read but they can be read when downloaded.

**Conclusions**

-Are the conclusions supported by the data presented?

-Are the limitations of analysis clearly described?

-Do the authors discuss how these data can be helpful to advance our understanding of the topic under study?

-Is public health relevance addressed?

Reviewer #1: This systematic review is of interest to the general public because it provides records and endemicity data to understand the extend of NTDs in RoC. For example, the information concerning the presence of loasis in areas endemic for onchocerciasis or Lymphatic Filariasis, and the potential for severe adverse events is critical. The geographical distribution of the diseases provides valuable information that the national NTD programs could use. Research can use the data for modeling. 

Some statements (line 638) are more speculative since no study was done to ascertain that the disease was imported from other countries. In addition, it is unlikely that observations in 1953 (line 638) would validate a statement from 1920 (line 635). The first sentence of the conclusion is not in line with the finding, which notes that Trachoma was not considered endemic at the national level. Although the last section of the paper eludes to some significant limitations that the authors should bring up, there is no limitation section.

Reviewer #2: The authors propose that this information would be useful for all those planning to work on NTDs in Congo and it is an excellent summary. There are suggestions for filling some of the gaps, particularly where better diagnostics could help. This would be a very useful public health resource.

**Editorial and Data Presentation Modifications?**

Reviewer #1: There are some editorial modification that need consideration:

Line 29: The qualification “surprising” might not be accurate because NTDs are not well known and are widely documented. This mention of “surprise” is unnecessary and should be removed

Line 33 -34: Like “LF” the acronym “STH” should be included because they are used later in the text 

Line 42: Please clarify what “LF cases” means. It is referring the LF antigenemia or LF morbidity cases/ Please be more specific.

Line 45: Please specify if the prevalence of trachoma in the Likouala department. Was the prevalence below 5% TF?

Line: 46-47 – The sentence - However, the overall number of publications was low for the period under study and remained so in recent years – it does not fit well in the conclusion section. It should be better moved up in the finding section.

Line 80: - Neglected Tropical Diseases (NTDs) are a group of primarily communicable and parasitic tropical diseases - Please note here that Trachoma is not a parasitic disease. It is bacterial. The sentence should be amended. 

Line: 88: “symptoms” and “clinical signs” look redundant; please stick with one of them. 

Line 107: The figures here do not add up. The total of the numbers shows less than 100,300 deaths.

Line 140: Please change “on” by “about”

Line 145: Method: It is clear if the non-PC NTDs were excluded from the method. Please clarify.

Line 147: Please specify the starting date and time

Line: 175: The sentence is in the past. Please correct

Line 189: The method section does not mention inclusion criteria

Line 202: “and then” please remove the “and”

Line 207 and 210: Please spell out the acronyms ‘PNLO” and “PNLSCH” as they appear for the first time in the manuscript

Line 2010-211: The sentence is confusing; please rephrase

Line 236: Please remove “as well as” and place by ‘and”

Line 264: Please insert a reference 

Line 267: Please provide specific data instead of “very low prevalence”

Line 276: “PNod” please spell out. PNod is an unusual acronym for nodule prevalence; please change it throughout the manuscript

Line 276: “PMF” – What does this stand for? Please spell out

Line 301: Please change “of” by “with”

Line 329: Please replace “serious’ by “severe” for adverse events

Line 333: Please provide a reference – Please note that high level loasis microfilaria might still be a significant risk of severe adverse events for onchocerciasis MDA 

Line 365 and 368: Please insert references

Line 396: Please clarify if this is “therapeutic coverage”

Line 423: Please remove “possible” before ‘presence’

Line 456, 470, and 473: Please include references

Line 499: HD and IU are used interchangeably; please stick with one.

Line 504: Table 4 - This table is not indispensable. The treatment schedule is already described in the text. 

Line 524: Strongyloidiasis is not commonly cited as part of the main STH. The STH here should stick to hookworm, Ascaris and Trichiuris.

Line 631-632: Please clarify that this guidance applies only to the inception of a control program from baseline mapping

Line 660 – 663: This sentence looks like an expert opinion made at in 1964. It might be more relevant and useful to provide data and evidence.

Line 637: When was the schistosomiasis program created? Please clarify.

Line 731: Please add that blinding trachoma is targeted for elimination by WHO.

Line 749: Please remove “included” in the title of table 8. This mention is not necessary.

Line 782-783: Please clarify if “low number of cases” refers to LF antigenemia or morbidity cases.

Line 793: The authors should note that there has been no nationwide mapping of trachoma in RoC. No desk review and no trachoma rapid assessment were done in suspected areas.

Lines 799 to 811: The paragraph should be moved up next to the section of the disease description.

Lines 812 -826: This section should be called “limitation of the review”. 

Line 823: It should be noted that now serological tests (OV16) are more sensitive for the mapping and might provide different endemicity prospects than the nodules and microfilaremia used in the past.

Figure 1 to 9 are titles missing

Reviewer #2: This is a long article and for those unfamiliar with some of the subjects in might be good to have a list of abbreviations to avoid searching back through the document to understand it.

With regard to Onchocerciasis different authors use different descriptive terms. I think PNod, presumably Nodule Prevalence and PMF, microfilaria prevalence, not density or CMFL could be more clearly defined.

Loa has been mentioned, it might also have been good to add an old reference to spontaneous Loa encephalopathy in one of the places you have discussed this.

In line 613 I would add the risk of HIV infection in female urogenital shistosomiasis.

**Summary and General Comments**

Reviewer #1: The paper should provide a summary of NTD prevalence and epidemiological situation. But, Instead, it focuses on the availability of publications and literature. It does not focus enough on the NTD situation.

Reviewer #2: This is a well researched article and contains much information and very useful for anyone working in the Republic of Congo. I think it will prove to be a useful resource.

I think this would be of interest to those working in Congo but not of a general interest to warrant special media coverage.

PLOS authors have the option to publish the peer review history of their article (what does this mean?). If published, this will include your full peer review and any attached files.

Reviewer #1: No

Reviewer #2: No

Figure Files:

Data Requirements:

Reproducibility:

References

---

## [Decision Letter · Decision Letter 1]

5 Apr 2022

Dear Dr Chesnais,

Thank you very much for submitting your manuscript "Historical overview and geographical distribution of Neglected Tropical Diseases amenable to preventive chemotherapy in the Republic of the Congo: a systematic review" for consideration at PLOS Neglected Tropical Diseases. As with all papers reviewed by the journal, your manuscript was reviewed by members of the editorial board and by several independent reviewers. The reviewers appreciated the attention to an important topic. Based on the reviews, we are likely to accept this manuscript for publication, providing that you modify the manuscript according to the review recommendations. 

The article presents an overview of the distribution of neglected tropical diseases amenable to preventive chemotherapy in Congo.

Sincerely,

Amadou Garba

Associate Editor

Justin Remais

Deputy Editor

The article presents an overview of the distribution of neglected tropical diseases amenable to preventive chemotherapy in Congo.

Reviewer's Responses to Questions

**Key Review Criteria Required for Acceptance?**

**Methods**

-Are the objectives of the study clearly articulated with a clear testable hypothesis stated?

-Is the study design appropriate to address the stated objectives?

-Is the population clearly described and appropriate for the hypothesis being tested?

-Is the sample size sufficient to ensure adequate power to address the hypothesis being tested?

-Were correct statistical analysis used to support conclusions?

-Are there concerns about ethical or regulatory requirements being met?

Reviewer #1: (No Response)

Reviewer #2: The objectives are stated as a review article and in this is what has been done. More actual data inn this revision has improved the paper.

**Results**

-Does the analysis presented match the analysis plan?

-Are the results clearly and completely presented?

-Are the figures (Tables, Images) of sufficient quality for clarity?

Reviewer #1: (No Response)

Reviewer #2: Yes

**Conclusions**

-Are the conclusions supported by the data presented?

-Are the limitations of analysis clearly described?

-Do the authors discuss how these data can be helpful to advance our understanding of the topic under study?

-Is public health relevance addressed?

Reviewer #1: (No Response)

Reviewer #2: Yes

**Editorial and Data Presentation Modifications?**

Reviewer #1: (No Response)

Reviewer #2: A few minor comments: 

Line 383: The definition of Geographical Coverage is incorrect although correctly used further in the document. For APOC the Geographical Coverage is related to Meso and Hyper endemic foci. I would suggest changing the section in brackets to (During APOC GC was the proportion….. 

Line 435, 440 etc, it is a bit difficult to understand the administrative system and how the public health framework fits into all this. Are health districts based on ADs. It would help to understand if this could be explained.

Line 607. I think this is usually described as a “point of care” test.

Line 714. Trachoma is worldwide and still present in many Asian countries. The “first” infection may be with the under 5s but as stated further it is the repeated infection that creates the problem. 

Line 719: Initially the objective was to eliminate “blinding trachoma” but now is stated as eliminating trachoma as a “public health problem”

Line 724: Treatment for very small children is based on tetracycline eye ointment.

Line 817 : Could be the key “to fix” not “for fix”

**Summary and General Comments**

Reviewer #1: Here below are my comments for the second review and for the responses to my first observations: 

1) Line 41-42 Abstract - I am still struggling with the statement of the authors noting that "trachoma is not endemic at the national level". Trachoma is a focal disease and therefore if one or several districts are endemic (TF> 5%) then epidemiologically the country is endemic for Trachoma. I’d suggest rephrasing to read as follows: “Trachoma is only endemic in the department of Likouala (prevalence>5%), further mapping is essential to properly assess the burden of the disease in the country”

2) Response Line 631-632: For Schistosomiasis, the WHO 2011 Guidelines recommend that AFTER 5-6 years of treatment, in high and moderate setting, preventive chemotherapy be conducted once every two years (for areas with prev 10-50%) or two times a year (for areas with prevalence still high Prev>50%)

Reviewer #2: A very thorough document that would be of use to someone new to NTDs in the Republic of Congo

PLOS authors have the option to publish the peer review history of their article (what does this mean?). If published, this will include your full peer review and any attached files.

Reviewer #1: No

Reviewer #2: No

Figure Files:

Data Requirements:

Reproducibility:

References

---

## [Editor Report · Decision Letter 2]

3 Jun 2022

Dear Dr Chesnais,

We are pleased to inform you that your manuscript 'Historical overview and geographical distribution of Neglected Tropical Diseases amenable to preventive chemotherapy in the Republic of the Congo: a systematic review' has been provisionally accepted for publication in PLOS Neglected Tropical Diseases.

Best regards,

Amadou Garba

Associate Editor

Justin Remais

Deputy Editor

---

## [Editor Report · Acceptance letter]

7 Jul 2022

Dear Dr Chesnais,

We are delighted to inform you that your manuscript, "Historical overview and geographical distribution of Neglected Tropical Diseases amenable to preventive chemotherapy in the Republic of the Congo: a systematic review," has been formally accepted for publication in PLOS Neglected Tropical Diseases.

Best regards,

Shaden Kamhawi

co-Editor-in-Chief

Paul Brindley

co-Editor-in-Chief
